



# Cutting peatland CO₂ emissions with rewetting measures

*Jim Boonman[1], Mariet M. Hefting [2], Corine J. A. van Huissteden[1], Merit van den Berg[1], Jakobus van Huissteden[1], Gilles Erkens[3,4], Roel Melman[4], Ype van der Velde[1]*

[1]Earth Sciences, Vrije Universiteit Amsterdam, Amsterdam, 1081 HV, The Netherlands
[2]Institute of Environmental Biology, Utrecht University, Utrecht, 3508 TB, The Netherlands
[3]Department of Physical Geography, Utrecht University, Utrecht, 3508 TC, The Netherlands.
[4]Deltares Research Institute, Utrecht, 3584 BK, the Netherlands

*Correspondence to*: Jim Boonman (j.boonman@vu.nl)

**Abstract.** Peat decomposition in managed peatlands is responsible for a decrease of 0.52 GtC yr$^{-1}$ in global carbon stock and is strongly linked to drainage to improve the agricultural bearing capacity, which increases aeration of the soil. Microbial aerobic decomposition is responsible for the bulk of the net $CO_2$ emission from the soil and could be reduced by wetting efforts or minimizing drainage. However, the effects of rewetting efforts on microbial respiration rate are largely unknown. We aimed to obtain more insight in these rewetting effects and measured them for 1 year for two dairy farming peatlands where submerged drainage subsurface irrigation (SDSI) was tested against a control situation. With a modelling approach, we explored the effects of rewetting under different weather conditions, water management strategies (raising ditch water levels and SDSI) and hydrological settings. We introduced a methodology to estimate potential aerobic microbial respiration rate as measure for peat decomposition in managed peatlands, based on potential respiration rate curves for soil temperature and water filled pore space (WFPS). Rewetting with SDSI resulted in higher summer groundwater levels, soil temperatures and WFPS. SDSI reduced net ecosystem production (NEP) with 1.27 ± 0.39 kg $CO_2$ m$^{-2}$ yr$^{-1}$ (83%) and 0.78 ± 0.37 kg $CO_2$ m$^{-2}$ yr$^{-1}$ (35%) for Assendelft and Vlist respectively. With the process based modelling approach we found that raising ditch water levels always reduces peat respiration rates. Furthermore, we found that the application of SDSI reduces yearly peat respiration rates in environments in a dry year and/or with downward hydrological fluxes, and increases peat respiration rates in a wet year and/or when upward groundwater fluxes are present. Moreover, combining SDSI with high ditch water levels or pressurizing SDSI systems will further reduce peat respiration rates. We highly recommend to use a process-based approach based on temperature and WFPS soil conditions to determine effectivities of rewetting efforts over empirical relationships between average groundwater level and NEP. Such a more process based approach allows to distinguish between groundwater levels raised by SDSI and ditch water levels. When this is not possible, we recommend using mean summer groundwater level instead of mean annual groundwater level as a proxy to estimate NEP. Such relations between mean groundwater levels and NEP need to be corrected for situations with SDSI.

## 1    Introduction

As a result of thousands of years of accumulation of organic material under waterlogged anoxic conditions, peat soils have formed extensively since the Last Glacial Maximum (Yu et al., 2010). Peat soils are an important carbon stock in the global



carbon cycle. Despite their relatively small global surface area of 3% (Yu et al., 2010; Leifeld & Menichetti, 2018; Friedlingstein et al., 2019), peat soils contain 600 GtC of carbon, equivalent to two third of the atmospheric carbon (Leifeld & Menichetti, 2018). Reclamation of the peatlands for forestry and agriculture, and mining peatlands for potting soil and fuel is responsible for a significant contribution to global warming. Currently, 13% of the global peat surface area is decomposing, which is responsible for an estimated annual decrease of 0.52 GtC in global carbon stock (Leifeld & Menichetti, 2018).

Peat decomposition in managed peatlands is strongly linked to drainage, often for agricultural purposes (e.g. Erkens et al., 2016; Tiemeyer et al., 2020; Evans et al., 2021). Drainage, both by ditching and subsurface tube drain pipes, is a necessity to create the bearing capacity for cattle and agricultural machines (Erkens et al., 2016). However, lowering groundwater levels increases the oxygen intrusion into the soil. Especially during dry summer periods, groundwater levels often drop to 0.80 m below the surface or more (e.g. Querner et al., 2012). Aerobic soil conditions that arise with increased oxygen intrusion stimulate microbial decomposition, a process which is responsible for the bulk of emission of $CO_2$ from the soil (old carbon material) and land subsidence (Dolman et al., 2019 p. 250; Tiemeyer et al., 2020). The obvious approach to reduce microbial peat decomposition is to minimize oxygen intrusion during the summer period. Raising the surface water levels during summer and the application of submerged drain subsurface irrigation systems (SDSI) target to decrease the thickness of the unsaturated zone and the active period for aerobic microbial decomposition. At the same time, SDSI functions as a drainage system when high groundwater tables occur, which causes dry enough conditions for agricultural practice in early spring (Kechavarzi et al., 2007; Querner et al., 2012; Weideveld et al., 2021).

Field research has shown that peatland groundwater level can be used as a proxy for ecosystem respiration and $CO_2$ fluxes (Fritz et al., 2017; Tiemeyer et al., 2020; Evans et al., 2021). However, some studies report dissonant results for particular field sites in which groundwater levels do not explain measured $CO_2$ fluxes, possibly due to peat decomposition being moisture limited due to extreme drought (Parmentier et al., 2009; Tiemeyer et al., 2016; Sieber et al., 2021). Within the unsaturated soil, soil temperature and moisture, often defined as water filled pore space (WFPS), are the main physical drivers of aerobic microbial respiration rate (Mäkiranta et al., 2009; Kechavarzi et al., 2010; Moyano et al., 2013; Bader et al., 2018). Aerobic microbial respiration rate is enhanced (often exponentially) as the soil temperature increases. When the WFPS decreases, the soil becomes aerated and potential microbial respiration rate increases. However, when the WFPS reduces further, microbial growth decreases due to lack of water as a solute exchange medium and a higher energy investment to obtain osmotic equilibrium (Moyano et al., 2013). $CO_2$ emission peaks of aerobic microbial respiration rate in peat have been established between 56-92% WFPS (Säurich et al., 2019).

Field measurements in agricultural peat meadows have shown that high surface water (ditch) levels alone will not be able to support high groundwater levels in the center of the peat meadows during summer (Hooghoudt, 1936; Jansen et al., 2007; Kechavarzi et al., 2007). This is attributed to the high evaporation and infiltration fluxes and the low horizontal conductivity of the peat soil. In the SWAMPS project (Sieber et al., 2021), it has been argued that increases in ditch water levels do not



necessarily lead to lower $CO_2$ emissions. However, there are also studies that describe a strong dependency of land subsidence (as measure for peat decomposition) on ditch water level (van den Akker et al., 2008).

Submerged drain subsurface irrigation (SDSI) has been introduced as a management measure to reduce groundwater table fluctuations and concomitantly reduce peat oxidation rates (Best & Jacobs, 1997; Kechavarzi et al., 2007). However, the effects

of SDSI on peat decomposition have not thoroughly been researched. Field measurements on the application of SDSI systems have shown land subsidence reductions (as measure for peat decomposition reduction) by 50% (van den Akker et al., 2008), but in some cases no effects in net ecosystem production (NEP) could be discovered (Weideveld et al., 2021), or effects differed strongly with reductions or increases in $CO_2$ emission from year to year (Sieber et al., 2021). SDSI consists of regularly spaced (4- 10 m) submerged drain tubes in the soil at 0.4 – 0.7 m depth connected to a (surface) water system. During dry periods

with high evapotranspiration fluxes, the tubes supply water into the soil. During periods with water excess, the drain tubes discharge water out of the soil. SDSI thus overcomes the problem of limited control of the surface water system on the meadow groundwater levels and strongly limits yearly groundwater level fluctuations. However, SDSI infiltration of warm surface waters can potentially induce adverse effects. For example, SDSI is prone to creating a zone with constant optimal soil moisture conditions close to the warm surface. In theory, SDSI can thus also increase microbial respiration rate. In the study of

Weideveld et al., 2021 the upper soil zone (60 cm), that remained unsaturated after SDSI was applied, was suspected to be responsible for the bulk of respiration as compared to the deeper soil layers that were saturated by SDSI. It is therefore that the authors think that SDSI had no or little effect on peat respiration. This divergence in observed effectivity estimations of rewetting measures indicates that a better process understanding is needed to assess the effectiveness of different water management strategies to reduce greenhouse gas (GHG) emissions for various peat types, landscape settings and climate

conditions.

Models are important tools to assess nationwide GHG emissions from peat areas (IPCC, 2014). The most elemental GHG-model is based on land use and climate specific emission factors to calculate nation-wide GHG-emissions, like the tier 1 IPCC approach (IPCC, 2014). Within more complex methods, emission factors are related to long term average groundwater levels

(tier 2) (Bechtold et al., 2014; Arets et al., 2020; Tiemeyer et al., 2020) or are derived from elaborate approaches that involve comprehensive understanding and validated representations of carbon dynamics using field measurements and modelling (tier 3) (IPCC, 2014). The latter approach is not yet used on country-scale (e.g. Arets et al., 2020; Tiemeyer et al., 2020) and most recent studies advocate to use water table depth peat decomposition relationships for nationwide GHG emissions. However, studies that have assessed such relationships did not involve sites with SDSI groundwater table manipulation. In this study we

aim to find the effectivity (in terms of reducing peat decomposition) of water management strategies (raising ditch water level and the application of SDSI) for different hydrological environments and different meteorological conditions. We present detailed measurements for 1 year of in-situ $CO_2$ fluxes for 2 pilot sites in the Netherlands, each site including a meadow with and without SDSI. Additionally, we modelled the groundwater flow for these sites and derived an estimated




potential microbial respiration rate from the simulated unsaturated soil temperature and WFPS: two main drivers of peat

decomposition. This allowed us to use the model to cautiously explore and identify the hydrological and climatic conditions

at which peat decomposition can be reduced by surface water levels manipulation and/or installing SDSI.

## 2     Methodology

Here we present the measurement setup belonging to the first year of measurements of a national monitoring effort to quantify

the effects of SDSI systems on groundwater level, soil temperature, WFPS and carbon emissions of peat meadows used for

dairy farming. A cross-sectional 2D hydrological model and the concept of "potential microbial respiration rate" are introduced

to simulate the field sites and explore the expected effectivity of SDSI and raising surface water levels under different

meteorology, and regional hydrological seepage settings. This model setup was validated on the available measurements (Fig.

1).

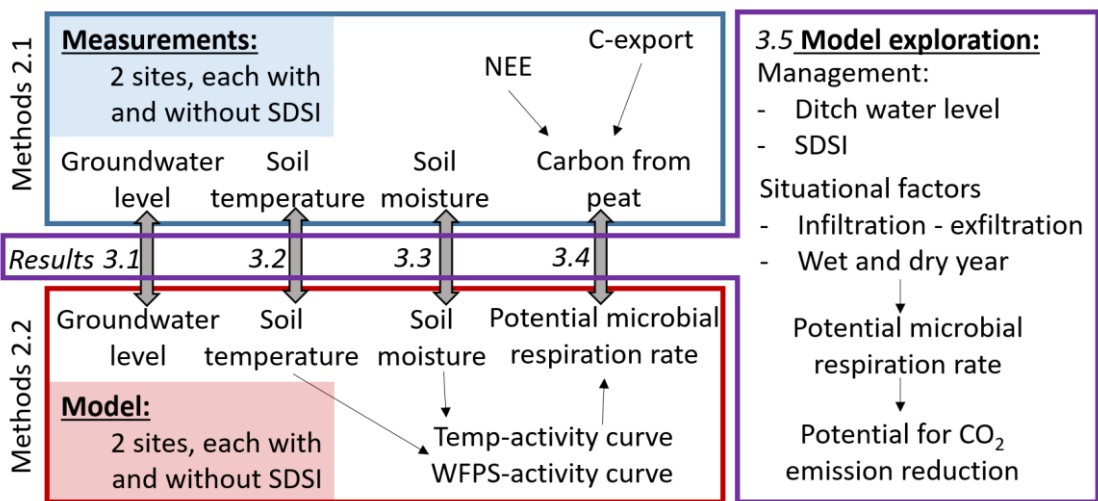


**Figure 1 – An overview of the research methodology.**



## 2.1 Field data

Measurements of groundwater level, soil temperature and WFPS were done at two locations consisting of managed meso-, eutrophic fen peatlands in the Netherlands: Assendelft and Vlist (Fig. 2). Both research locations are in agricultural use and consist of two parcels that are monitored: one without (control) and one with SDSI.

### 2.1.1 Site descriptions
#### 2.1.1.1 Assendelft


The research site in Assendelft (ASD), located in the province Noord-Holland, is a managed eutrophic peatland used for dairy farming (Fig. 2a, b). The clayey degraded moor topsoil (0-30 cm) covers a slightly clayey degraded moor horizon (30-40 cm). Mildly degraded reed/sedge peat is present underneath the degraded peat layers (40-200 cm), followed by marine clay deposits. Reduced hydrogen sulfide in the reduced soil layers suggests that the peatland is exposed to upward seepage which is confirmed

by groundwater observation wells that consist of a higher hydraulic head in the deep (>10 m depth) subsurface as compared to the topsoil (< 1.5 m depth) hydraulic head. In the eastern meadow, no SDSI system is present. In 2017, a pressurized SDSI system was constructed in the western meadow. These systems function the same as SDSI discussed earlier, but include a pressure container that is used to manipulate the pressure in the drains independently from ditch water levels. The drains are located 50-70 cm below the surface and spaced at a distance of 4 m. In the summer, drain pressure varied between +5 and -40

cm from the surface with a mean of -17 cm. Summer ditch water levels are kept at 40 cm from the surface. The two parcels that are monitored have a total width of 180 m, with trenches of 30-40 cm deep spaced 12 to 19 m.

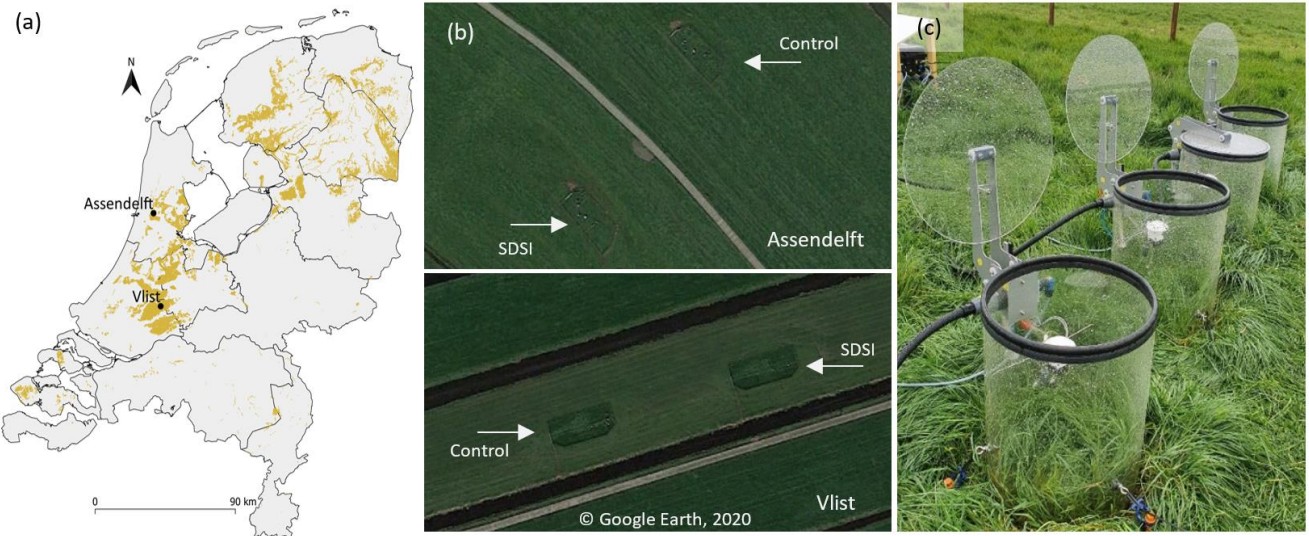

**Figure 2 – (a) Peat soils in the Netherlands (highlighted) with the research locations Assendelft and Vlist on national scale (Basisregistratie Ondergrond, 2018; Esri Nederland, 2021), (b) the research locations seen from 500 m altitude with satellite imagery**
**(the borders of the measuring plots are recognizable by the fences) and (c) the automated transparent chamber systems in Assendelft.**





#### 2.1.1.2 Vlist

The research location in Vlist (VLI), provice Zuid-Holland, is a comparable managed fen peatland used for dairy farming (Fig.
2a, b). The clay topsoil (0-40 cm) covers a degraded peat layer (40-55 cm). Meso, -eutrophic woody, sedge and reed peat is
present underneath the degraded horizon (55-290 cm), including a clay layer that is thicker when moving towards river Vlist
in the west (5-40 cm thick, around a depth of 220 cm). Underneath the peat, a clay layer (1-2 m thick) was found. In 2011,
SDSI systems were installed at the research location, with a depth of 70 cm and a drain spacing of 6 m. The SDSI systems are
oriented perpendicular to and directly connected to the ditch. Ditch summer water levels are aimed to be at 50 cm below the
surface. The width of the parcel is around 35 m with a trench of 20-30 cm deep in the middle.

#### 2.1.2 Groundwater observation wells

To assess the effect of SDSI on the groundwater table observation wells were used. The spacing between the groundwater
wells and the ditch was 60 m and 12 m for Assendelft and Vlist respectively. The depth of the groundwater observation well
filters was 0.3-1.3 m below surface and the wells were funded in deeper soil layers (> 3 m) to prevent influences of vertical
soil movement on the measurements. Bottom well pressure was measured and automatically corrected for atmospheric pressure
with Ellitrack-D logging equipment (Leiderdorp Instruments, the Netherlands).

#### 2.1.3 Temperature & WFPS

The effect of SDSI on soil temperature (0.1 - 1.2 m depth) and WFPS was measured with two Sentek Drill & Drop (Sentek
Sensor Technologies, Australia) probes at a depth interval of 0.1 m up to 1.2 m below surface on each meadow.
At the SDSI plots, WFPS was measured from the drains at half and a quarter of the distance between the drains. Raw sensor
measurements (scaled frequency) were corrected for soil temperature and converted to soil moisture percentages with a linear
calibration curve that was derived for a similar peatland in the Netherlands. The curve had a slope of 0.74 and an intercept of
-0.02. WFPS was calculated as the soil moisture relative to maximum soil moisture at that particular depth. Uncertainty in
WFPS estimations was likely to be high as the calibration curve was not derived for the specific field locations.

#### 2.1.4 $CO_2$ fluxes, harvesting and carbon budget

Automated transparent chamber systems were deployed on the research sites (Fig. 2c) to measure the ecosystem $CO_2$ fluxes
that were used to determine nocturnal ecosystem respiration ($R_{eco}$) and net ecosystem exchange (NEE). Each system consisted
of four chambers that had a height of 0.5 m and a diameter of 0.4 m. Every 15 minutes, one measuring cycle was executed in
which every chamber was closed by a lid independently for 3 minutes. Air from the closed chamber was drawn into the LI-
850 $CO_2$ gas analyzer (LI-COR, USA), which determined $CO_2$ and $H_2O$ concentrations with a measuring interval of 2 seconds.





By fitting a robust linear model to the first 60 seconds after lid closing, the $CO_2$ flux during chamber closure was calculated. Fluxes were converted from volume to weight by using chamber temperature, gas-cell pressure and chamber height. Measurements were corrected for water vapor and chamber cycles were checked for air leaking and fan malfunctioning using
$R^2$ values of the fitting process. Chamber systems were relocated every two weeks to minimize the effect of the chamber 'micro-climate' on the vegetation and soil. To assess $R_{eco}$, nocturnal chamber flux measurements were selected -by using sunrise, sunset and chamber closure time data- and subsequently averaged on each date.

Crop (*Lolium perenne*) harvesting was done five times throughout the growing season, simulating the farmer's harvesting practice. Samples inside and around the chambers were taken in quadruplo. The dry weight and carbon content of each sample
were determined in the lab. Subsequently, carbon yield per area was calculated. The carbon yield per area (C-export) was corrected for the difference between inside and outside chamber crop growth for each measuring plot (supplementary information, S3). Standard deviations for each harvesting event were based on the differences between the samples within the same sample group.

The carbon budget (NEP, net ecosystem production) of the research sites was calculated by summing NEE and C-export
determined within the measuring period, similar to the approach presented by Görres et al., 2014. We selected a one-year measuring period ranging from the first of April 2020 until the first of April 2021. NEE was calculated by summing daily averages of chamber measurements. Standard deviations of the daily averages were calculated with daily NEE means of each chamber. Missing values were filled with 20-day averages around the period without measuring data, with a maximum missing consecutive period of 15 days in autumn for one site (Vlist control). Near the end of 2020 (during low carbon fluxes) the
systems were calibrated. NEE data were interpolated for April 2020 when automatic flux chambers at site Vlist were not yet deployed (supplementary information, S3). The maximum amount of gap-filled chamber data was 16% (Vlist control). The total harvested carbon yield consisted of the sum of harvests and was expressed in mass units of $CO_2$ per area. The total standard deviation of yearly NEE and C-export was calculated by taking the square root of the summed squared standard deviations of the averages that were used within the calculation. We assumed that the carbon storage in the crop at the beginning of the
measuring year equaled the carbon storage at the end of the measuring year. Crop fertilization was done with inorganic fertilizers so that no carbon was added to the soil.

The effectivity of SDSI was determined by comparing NEP of the control site with the SDSI site for each location. The effectivity was expressed as an absolute number and percentage of the reduction in the NEP.

## 2.2   Model setup

We simulated a typical dairy farming grassland on sedge peat for the Netherlands exposed to various natural and artificial (boundary) conditions: soil saturated hydraulic conductivity, the presence of up- or downward flow, meteorology, ditch water level and the application of SDSI systems. The model runs differed in their ditch water level, seepage flux, hydrological conductivity, weather years and application of SDSI. Daily weather and temperature and parcel width were kept constant for
all model runs. The goal of this model was not to exactly reproduce the observation sites, but to capture the main effects of





surface water level and SDSI on soil temperature and soil moisture. When these dynamics are adequately described the model can be used to cautiously explore the effects of SDSI and water level management on yearly average peat respiration rates.

### 2.2.1    Hydrological model


We used the hydrological 2D finite element method model HYDRUS (3.02) (Simunek, J., Sejna, M., & Van Genuchten, 2014). The HYDRUS software is proven to reliably simulate the hydrology in the unsaturated zone and the soil temperature dynamics, which are key features in this research, using the Richard's equation and conduction and convection equations.

The model domain was designed to represent a typical cross section of a managed peatland in the Netherlands (Fig. 3). In the

Netherlands, parcel widths from ditch to ditch vary between 25-100 m. In this research a parcel was used with a standard width of 35 m. We used a soil depth of 3 m and a shallow surface drainage trench in the middle of the parcel with a depth and width of 0.3 m and 0.9 m respectively. The infiltration/drainage ditch was 1.5 m deep and had a width of 1.5 m. Additionally, for SDSI model runs drains were introduced in the domain at a typical depth and distance interval of 0.7 m and 5.5 m respectively. The drains had a diameter of 0.07 m and were surrounded by 5 cm coarse sand. For computation purposes only half of the

cross-section has been modelled as it has been assumed that no groundwater would flow across the middle of the parcel. Element size was smaller at the top of the model (0.05m) and increased stepwise to 0.3 m at 0.8 m depth. The total amount of nodes and 2D-elements was around 10000 and 20000 respectively. Nodes neighboring boundary mesh points were refined if the distance ratio was too high (F = 1.3).

Soil properties were set up with the single porosity van Genuchten-Mualem hydraulic model (van Genuchten, 1980). Since

peat soils vary widely in hydraulic properties, we chose to represent sedge peat soil characteristics with optimized parameters from large dataset analyses (Liu et al., 2016; Heinen et al., 2018). The modelled soil consisted of a clayey peat topsoil (Heinen et al., 2018), a decomposed peat layer of 40 cm and a sedge peat layer (Liu et al., 2016). Further details and Van Genuchten-Mualem parameter values are presented in Table 1. Data availability was insufficient to incorporate the dual-porosity, heterogeneous and anisotropic nature of the peat soils


**Table 1 - Mualem - Van Genuchten parameters and saturated hydraullic conductivity for each soil layer**

| Soil type | depth [cm] | $\theta_r$ | $\theta_s$ | $\alpha$ [1/m] | n | $K_s$ [m d$^{-1}$] | $\tau$ |
|---|---|---|---|---|---|---|---|
| Clayey peat[2] | 0-20 | 0 | 0.765 | 2.05 | 1.151 | 0.1314 | 0 |
| Decomposed peat[1] | 20-60 | 0 | 0.77 | 1.4 | 1.16 | 0.0552 | 0.5 |
| Sedge peat[1] | >60 | 0 | 0.88 | 2.9 | 1.21 | 0.468 | 0.5 |

1. Liu et al., (2016), 2. Heinen et al., (2018)



Thermal conductivity parameters for simulating temperature were estimated according to the empirical model of Dissanayaka et al. (2013) and were dependent on solid and water content. The model is presented in eq. 1, in which $\sigma$ and $\theta$ represent solid and water content respectively. Default heat capacity values were used for water, organic matter and clastic material.

$$\lambda = 0.225 \cdot \sigma + 0.025 + 0.89 \cdot \lambda_{water} \cdot \theta \qquad (1)$$


Meteorological input data was acquired from the weather station Schiphol of the Royal Dutch Weather Agency (KNMI, 2020). Runoff was calculated using a 1D-HYDRUS model with a bottom pressure head boundary equal to ditch water levels plus 0.1 m to account for the forming of a concave phreatic surface. The water that was available for infiltration was reduced for time-steps during which the model runs did not converge. For model runs including submerged drain systems, the water that was

available for infiltration was equal to precipitation. Throughout the year, 90% of potential Makkink evaporation was assumed to be represented by transpiration, and 10% by soil evaporation (Allen, R. G. et al., 1998). On top of the soil profile, we used measured average soil temperatures at 5 cm depth from Marknesse, Flevoland, the Netherlands (KNMI, 2020). Daily average river water (Hagestein, the Netherlands) temperatures (50 cm depth) were assigned to ditch and drain water (Rijkswaterstaat, 2020). A constant temperature of 11 °C was assumed at the bottom boundary at 3 m depth.

Root water uptake was described using the Feddes (1978) model with parameters for grass. Crop growth and crop solute stress were not included. Rooting depth was set at 0.3 m, with maximum rooting intensity at 0.1 m (root distribution parameter $P_z$ was set at 3).

Initial pressure head conditions were set at -0.2 m from the surface and temperature initial conditions were set at 11°C. We did not use a warming-up period for the model, as we expected near surface water tables and consequentially low potential

microbial respiration rate during winter.

### 2.2.2   Peat decomposition model: potential respiration rate

To estimate potential microbial respiration rate of aerobic microbial communities in the unsaturated zone (further in this paper

referred to as potential respiration rate), we introduce temperature-activity ($A_T(n,t)$ [relative respiration rate during one day]) and WFPS-activity ($A_{WFPS}(n,t)$ [relative respiration rate during one day]) curves for potential respiration rate. These activity curves quantify the potential respiration rate at each temperature and WFPS relative to a reference value of temperature and WFPS.

Soil temperature is assumed to influence potential respiration rate according to the square root equation, as described in Eq. 2.

with fitted $T_{min}$ and $a$ of -10 °C and 0.05 respectively (Ratkowsky et al., 1983; Lloyd J., Taylor, 1994; Bååth, 2018). WFPS additionally determines potential respiration rate. The optimum curve for WFPS was selected during the optimization of our estimation of potential respiration rate as explained in the next section. The number of total days with reference conditions for aerobic activity per year were calculated following Eq. 3 and 4.





$$A_T(n,t) = [a\,(T(n,t) - T_{min})]^2 \qquad\qquad (2)$$

$$A_u(n) = \sum_{t=1}^{365} A_T(n,t) \cdot A_{WFPS}(n,t) \qquad\qquad (3)$$

$$Ap = \sum_{n=1}^{nodes} A_u(n)\,S(n)/L \qquad\qquad (4)$$

For each soil node (n), potential microbial respiration rate ($A_u$(n) [day]) was calculated (Fig. 3). By multiplying each node with its surface area ($S$ [$m^2$]) and dividing by the domain length ($L$ [$m$]) a field average potential respiration rate per area ($Ap$ [day

m yr$^{-1}$]) is obtained. This number can be interpreted as the potential respiration rate equal to that of 1 m$^2$ over a depth of 1 m exposed to the optimal WFPS and 20 °C for $Ap$ days in a year.

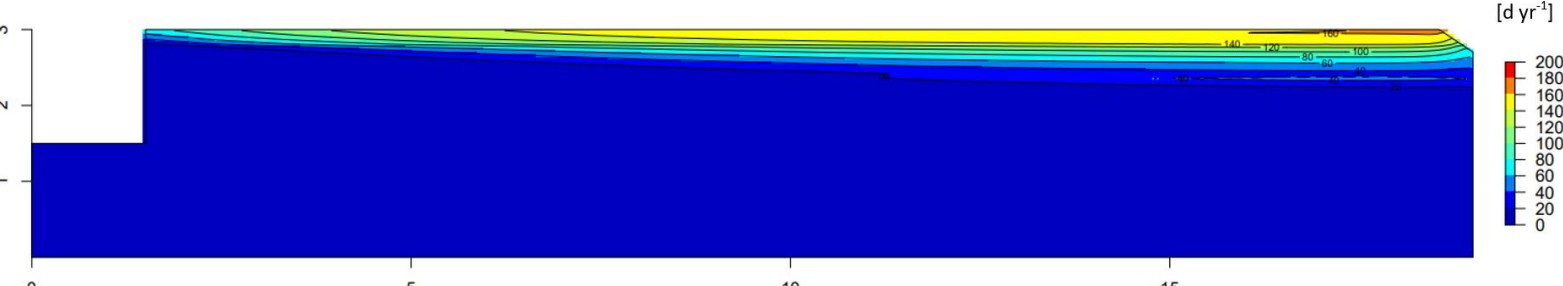

**Figure 3 – Model domain with estimated potential respiration rate [d yr$^{-1}$] for a model run.**

**2.2.3    WFPS potential respiration rate curve**

As limited scientific research is available on the relation between potential respiration rate and WFPS, we decided to test the performance of several WFPS-activity curve shapes (Fig. 4) in an ensemble analysis. We calculated potential respiration rate for the parcels in Assendelft and Vlist with all different WFPS relations, keeping the temperature relation with potential

reparation fixed. The $R_{eco}$ that was measured in the field partly consists of peat respiration driven by WFPS. Therefore, we assumed that the WFPS activity curve that produced the potential respiration rate that correlated best with the $R_{eco}$ most accurately described the relation between WFPS and potential respiration rate. The tested set of WFPS optimum curves (Fig. 4) was loosely based on the shape found by (Säurich et al., 2019).





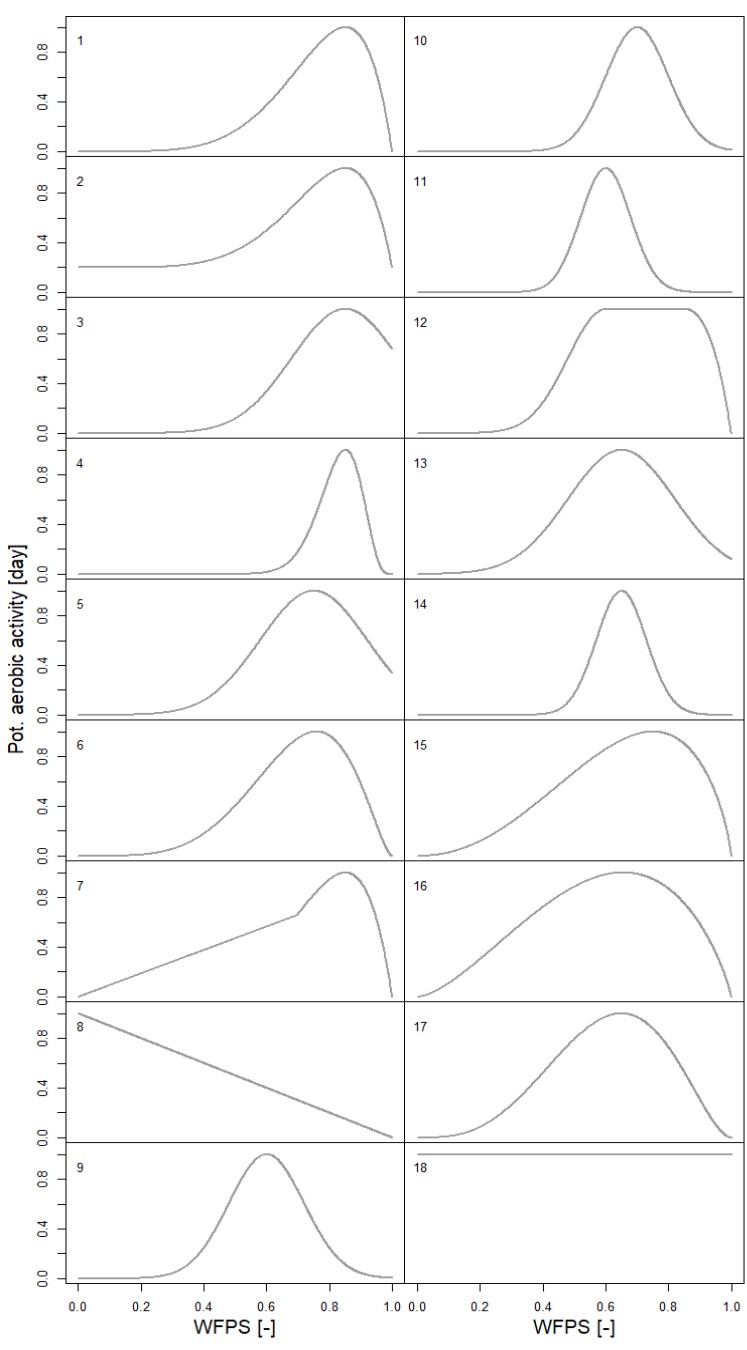

Figure 4 – The ensemble of WFPS optimum curves that was tested during model optimization.





### 2.2.4    Model variations

We simulated the field sites in Assendelft and Vlist with corresponding surface water levels and seepage fluxes with and
without SDSI for the measurement year 2020. To assess the performance of our model, we compared monthly means of
simulated groundwater level, soil temperature and WFPS with the measured monthly means of these variables. Additionally,
we related our measured estimations of NEP with simulated potential respiration rate to be able to compare our results with
recent literature.

In order to explore the effects of surface water level and SDSI on potential respiration rates we ran the model for a range of
surface water levels (0.2-0.6 m below surface) with and without SDSI, seepage conditions (1 mm d$^{-1}$ downward, 1 mm d$^{-1}$
upward), during both a climatically wet and a dry year (2012, 2018). The years 2012 and 2018 had a total precipitation balance
deficit of 14 (2012) and 296 mm (2018) from April to September, with 2018 belonging to the 5% most severely dry years since
1910 (KNMI, 2021). Ditch water levels in winter were varied from -0.3 m to -0.7 m, with 0.1m increased water levels during
the growing season from May 1$^{st}$ until October 1$^{st}$ as is common practice in the Netherlands. All model variations were run
with a hydraulic conductivity of the intact peat layer of 0.47, 0.23 and 0.12 m d$^{-1}$ (Liu et al., 2016).

### 3    Results

In this section, field measurements and model output are presented successively per theme and data are depicted in Fig. 5.

### 3.1    Groundwater level

Seasonal trends are clearly distinguishable in monthly averages of groundwater levels that were measured in the field (Fig.
5a), with higher and lower levels for SDSI research plots compared to control plots in summer and autumn/winter respectively.
SDSI establishes a more equal water table depth throughout the year as intended. The pressurized SDSI systems in Assendelft
maintained remarkably high groundwater levels during summer (ca. -0.2 m), as opposed to Vlist where the difference between
SDSI and control groundwater level seem confined to ca. 0.1m. Due to the greater spacing between ditch and observation well
in Assendelft we found lower water levels for the control plot as compared to Vlist, even though upward flow was present in
Assendelft.

The model captured the monthly dynamics of groundwater level reasonably well. Only the lower groundwater levels in Vlist
tend to be underestimated by 10 cm (Fig. 5b). Furthermore, we found that model runs underestimate the short-term variability
during summer that is observed at the field sites (supplementary information, S1).






**Figure 5 – Monthly averages of measured (a) groundwater level, (c) temperature and (e) WFPS that are compared with monthly modelled averages of (b) groundwater level, (d) temperature and (f) WFPS. Values for temperature and WFPS were averaged over a depth of 0.5 m. The legend in (a) applies for all panels.**

**3.2 Temperature**

In the field, maximal monthly soil temperatures of nearly 20°C in the top 0.5 m of the soil profile were measured in August for Vlist (Fig. 5c). The monthly difference between Assendelft and Vlist control was small with a maximum of 1.0°C. From June to August the topsoil is slowly warming, even though average air temperature was higher in June than in July. SDSI tends to cause higher monthly soil temperature levels (0.40°C) in the top 0.5 m of the soil profile in Assendelft in the summer. Over

the complete period, this heating effect was 0.20°C. In Vlist, SDSI tends to cause higher temperatures in this soil zone in all months except for August, with a summer maximum of 0.25°C for June and an autumn/winter maximum of 0.62°C in December. The average heating effect of SDSI on topsoil temperatures is 0.07°C and 0.21°C during summer and during the measured months in 2020 respectively.

Modelled and measured monthly topsoil temperature dynamics match well, but modelled temperatures were on average 1.45°C

higher than measured temperatures (Fig. 5d). The measured soil surface temperature data we present here was not yet available at the time of modelling. Hence we used a surface soil temperature dataset of a different location as top temperature boundary





condition which in hindsight had slightly too high temperatures explaining the model overestimation of the soil temperature. The simulated effect of SDSI on topsoil temperatures during summer was 0.72°C for Assendelft and 0.37°C for Vlist. Over the complete simulation year, the SDSI heating effect was 0.42°C and 0.26°C for Assendelft and Vlist respectively.


### 3.3    WFPS

The minimum monthly average WFPS in the top 0.5 m of the soil profile (Fig. 5e) was measured at the control site in Assendelft (0.76) during August 2020. WFPS tends to be higher during summer when SDSI is applied, especially with the pressurized SDSI systems in Assendelft. Here, the highest summer WFPS values are measured. Modelled WFPS agrees reasonably well

with measured WFPS but overestimates soil moisture for the control plot in Assendelft in September and underestimates (>10%) moisture values for the control plot in Vlist during summer drought (>10%) (Fig. 5f).

### 3.4    Approaching peat decomposition

### 3.4.1    Measured $R_{eco}$ and estimated potential respiration rate


Nocturnal $R_{eco}$ measurements clearly show seasonal trends with high $CO_2$ emissions in summer and low emissions in winter (Fig. 6a). Maximum $R_{eco}$ values were measured in August 2020 on the control parcel in Assendelft, approximating 70 g $CO_2$ $m^{-2}$ $d^{-1}$. In summer, $R_{eco}$ values measured on the control parcel were structurally higher than on the SDSI parcel on both research locations (Fig. 6b). In autumn, measured $R_{eco}$ was slightly higher in the SDSI parcel in Assendelft.

We compared measured nocturnal $R_{eco}$ with the estimated daily potential respiration rate calculated applying all the WFPS-activity curves in Fig. 6. Based on correlation between the calculated potential respiration rate and the nocturnal $CO_2$ fluxes, we found that WFPS-activity curves that include zero activity for a WFPS of 1, a general decrease in activity between WFPS of 0.65 and 1 (curves 8, 12,15 and 16) outperformed WFPS-activity curves that have an insensitive region between WFPS of 0.9-1 (curves 9-11) or those that have a clear peak activity above 0.7 WFPS (curves 1-4). We selected relation 16 (Fig. 4,

supplementary information, S2) for the analysis of our modelling output. The shape of this curve is matches most closely data presented in Säurich et al. (2019), who found optimal moisture conditions around a value of 0.65 WFPS. The curve is represented by a β distribution scaled to a maximum value of 1 with parameters α and β of 2.59 and 1.84 (resp.). For the research sites without and with SDSI in Assendelft and Vlist we found Pearson correlations of 0.73, 0.47, 0.61 and 0.56 (resp.) between potential respiration rate and measured nocturnal $R_{eco}$, with a mean of 0.59. Note that both the observed and simulated

WFPS did rarely drop below 65% for all 4 plots. Hence, effectively only the range of 0.65-1 WFPS of the relationships in Fig. 5 are compared.

The seasonal trends in measured $R_{eco}$ are present in the modelled respiration rate as well (Fig. 6). Model data suggests that temperature conditions are limiting respiration rate in spring and WFPS conditions limit respiration rate in autumn. Short term $R_{eco}$ variations are hardly present within estimations for potential respiration rate in the control fields, but are more evident in

simulations with SDSI. This is because the unsaturated zone in the SDSI fields are relatively close to saturation. The slope of





the WFPS respiration rate curve is steeper for wetter circumstances, so changes in WFPS affect respiration rate substantially. This effect is less important in the case of the unsaturated zone in the control field, as rainfall causes WFPS to shift around the optimum where the slope is marginal. As mentioned before, we have not subdivided $R_{eco}$ into peat and plant respiration. As plant respiration is responsible for the bulk of respiration, $R_{eco}$ differences between the control and SDSI field are less evident

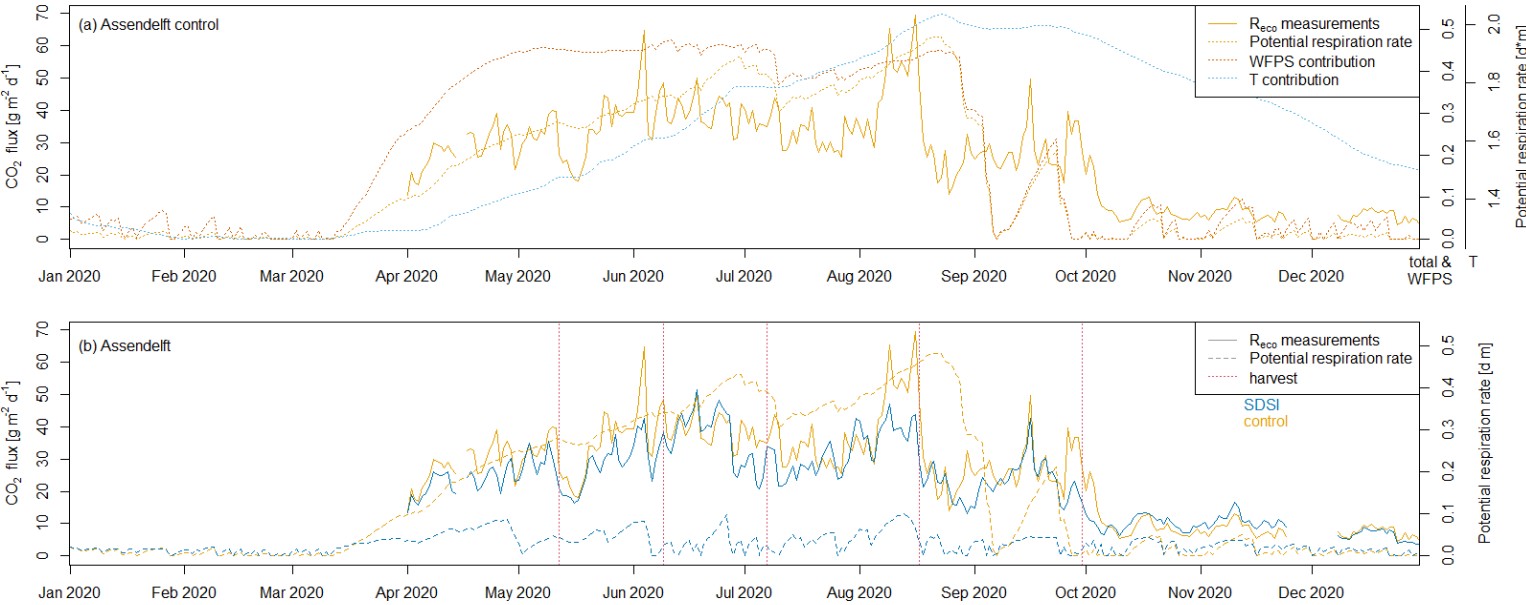

in field measurements than model peat respiration estimations.

**Figure 6 – (a) Measured nocturnal $R_{eco}$ and modelled respiration rate with the contribution of WFPS and temperature for the control site in Assendelft. (b) Measured $R_{eco}$ and modeled respiration rate in Assendelft for both the control and SDSI parcel. Line colors correspond with application of SDSI.**


### 3.4.2      NEP and water management strategy effectivity

On both research locations SDSI decreased C-export and NEE, leading to reductions in NEP (Table 4). We found the highest ($3.17 \pm 0.23$ kg $CO_2$ $m^{-2}$ $yr^{-1}$) and lowest ($2.42 \pm 0.28$ kg $CO_2$ $m^{-2}$ $yr^{-1}$) C-export for Assendelft control and Assendelft SDSI

respectively. The control plot in Vlist ($2.19 \pm 0.27$ kg $CO_2$ $m^{-2}$ $yr^{-1}$) emitted a higher amount of $CO_2$ emissions than the control plot in Assendelft ($1.53 \pm 0.25$ kg $CO_2$ $m^{-2}$ $yr^{-1}$). Our measurements show that the application of SDSI systems reduced $CO_2$ emissions for the dry year 2020 with $1.27 \pm 0.39$ kg $CO_2$ $m^{-2}$ $yr^{-1}$ ($83 \pm 25\%$) in Assendelft (pressurized SDSI) and with $0.78 \pm 0.37$ kg $CO_2$ $m^{-2}$ $yr^{-1}$ ($35 \pm 17\%$) in Vlist. Yield and NEE statistics can be found in supplementary information S3. Model calculations of simulated potential respiration rate (PRR) and effectivity of water management strategies agree with the

measured NEP and effectivity for both research locations (Table 4). Note that if we would account for the complete parcel by



looking at our model results from ditch to ditch (instead of using the results at the location representative for the measurement location in the middle of the parcel), the modelled effectivity would be 64% and 20% for Assendelft and Vlist respectively. We cautiously determined a relation ($R^2 = 0.81$) between measured NEP and modelled potential respiration rate (Fig. 7, Eq. 5) to be able to interpret and compare our model results with literature. To improve the reliability of this fitted relation, additional data from other locations and measuring years when they become available.

$$NEP = 0.033 * PRR - 0.13 \qquad (Eq. 5)$$

**Table 4 – Measured carbon balance in the growing season with corresponding effectivity of water management strategies and modelled potential respiration rate at the location representative for the measurement location with estimated effectivity of water management strategies.**

| | *ASD control* | *ASD SDSI* | *VLI control* | *VLI SDSI* |
|---|---|---|---|---|
| *NEE [kg $CO_2$ $m^{-2}$ $yr^{-1}$]* | -1.64 (0.10) | -2.17 (0.09) | -0.54 (0.10) | -1.15 (0.10) |
| *C-export [kg $CO_2$ $m^{-2}$ $yr^{-1}$]* | 3.17 (0.23) | 2.42 (0.28) | 2.74 (0.26) | 2.57 (0.23) |
| *NEP [kg $CO_2$ $m^{-2}$ $yr^{-1}$]* | 1.53 (0.25) | 0.26 (0.30) | 2.19 (0.27) | 1.42 (0.25) |
| *Measured effectivity [-]* | | 0.83 (0.25) | | 0.35 (0.17) |
| *Modelled PRR [d m $yr^{-1}$]* | 54.80 | 10.80 | 65.80 | 44.80 |
| *Modelled effectivity [-]* | | 0.80 | | 0.32 |

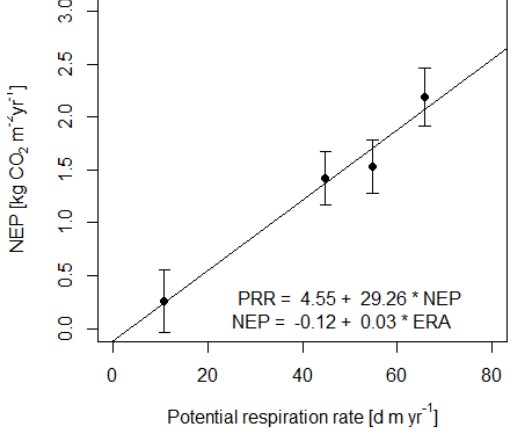

PRR = 4.55 + 29.26 * NEP
NEP = -0.12 + 0.03 * ERA

**Figure 7 – The relation between measured NEP [kg $CO_2$ $m^{-2}$ $yr^{-1}$] and potential microbial respiration rate (PRR[d m $yr^{-1}$]).**





### 3.5 Model exploration

Here we present the results of our model exploration study (depicted in Fig. 8). Model water balance errors were sufficiently low (<1% of precipitation) over the complete year for all the model simulations.

#### 3.5.1 Effects of meteorology and regional hydrology

We found lower potential respiration rate during wet years and in situations with upward flow across the bottom model boundary (Fig. 8a). Estimated potential respiration rate ranged between 3 and 81 d m yr$^{-1}$ (0.0 – 2.6 kg $CO_2$ m$^{-2}$ yr$^{-1}$). The highest amount of potential respiration rate was found for the dry model scenario with 1 mm d$^{-1}$ of downward seepage, a ditch summer water level of -60 cm and no SDSI. The lowest amount of estimated potential respiration rate was found for a wet model scenario with 1 mm d$^{-1}$ of upward seepage, a ditch summer water level of -20 cm and SDSI. Generally, a dry year caused the WFPS to drop at increased depths (0.8 m) as compared to a wet year (0.4 m), possibly affecting the pristine peat layer at greater depth (supplementary information, S4). In both wet and dry years the topsoil (20 cm) is most prone to a high potential respiration rate.

Changes in hydraulic conductivity of the intact sedge peat layer did not result in changes in potential respiration rate for SDSI model runs, and remained mostly confined to 10 d m yr$^{-1}$ for model runs without SDSI (supplementary information, S5).

#### 3.5.2 Effects of water management strategies

We determined the effects of water management strategies by comparing yearly potential respiration rate of model runs. The effect of ditch water level was calculated by comparing the results of each model run with the results of base scenario with a summer ditch water level of 50cm below surface (Fig. 8b). The effect of SDSI was calculated by comparing SDSI model runs with control model runs (Fig. 8b).

We find that elevating ditch water level consistently decreases potential respiration rate up to 18 d m yr$^{-1}$ (0.5 kg $CO_2$ m$^{-2}$ yr$^{-1}$) (Fig. 8b, supplementary information S5). Depending on the hydrological seepage setting, SDSI could increase and decrease potential respiration rate with +79% (12 m d yr$^{-1}$, 0.3 kg $CO_2$ m$^{-2}$ yr$^{-1}$) and -34% (24 m d yr$^{-1}$, 0.7 kg $CO_2$ m$^{-2}$ yr$^{-1}$) with a standard ditch water level of -50cm in summer. The application of SDSI is most effective during a dry year and with downward hydrological fluxes (Fig. 8b). In wet years SDSI is more likely to drain pores to field capacity, leading to a decrease in WFPS and increases in potential respiration rate as compared to control model runs. SDSI also increases potential respiration rate when upward hydrological flow is present across the bottom boundary. SDSI tends to have more positive effects on respiration rate when combined with high ditch water levels: up to -58% (30 m d yr$^{-1}$, 0.9 kg $CO_2$ m$^{-2}$ yr$^{-1}$) decrease in potential respiration rate (without including the decreases caused by ditch water level increment). This increase is due to the SDSI draining of the unsaturated zone in wet periods: the draining becomes less important when the depth of the unsaturated zone is minimized. We find that during dry years SDSI enhances water and heat transport to the topsoil resulting in stable zones with high potential





respiration rate close to the surface, although SDSI mostly tends to decrease the extent/depth of potential respiration rate in these years (supplementary information, S4).

**Figure 8 – (a) Dot graph of calculated potential respiration rate over the complete width of the model domain. Model runs are distinguished by (1) application of SDSI, (2) ditch water level from surface, (3) hydrological regime (1 mm d-1 upward to downward flow across the bottom boundary) and (4) meteorology (2018 for dry, 2012 for wet). (b) Effects of SDSI (above) and ditch water level (below) on respiration rate. Green dots represent a decrease in respiration rate. Effects of SDSI were determined by comparing similar model runs with and without SDSI. Effects of ditch water level were determined by comparing each scenario in which the**

**water level is 0.5 m below surface with the other scenarios in the same category on the x-axis.**

Reductions of $CO_2$ emissions during a dry year without upward seepage conditions could easily be overestimated by factor 2 (supplementary information, S5), which roughly seems to be a good indicator of differences between absolute reductions over the complete parcel and over the middle of the parcel (supplementary information, S5).





Sharpest declines in potential respiration rate are accomplished when deeper ditch water levels (-60 cm in summer) are raised
by 20 cm and with downward bottom fluxes (no SDSI, Fig. 9a). The spread of potential respiration rate within the different
seepage settings is diminished when SDSI is applied (Fig. 9b).

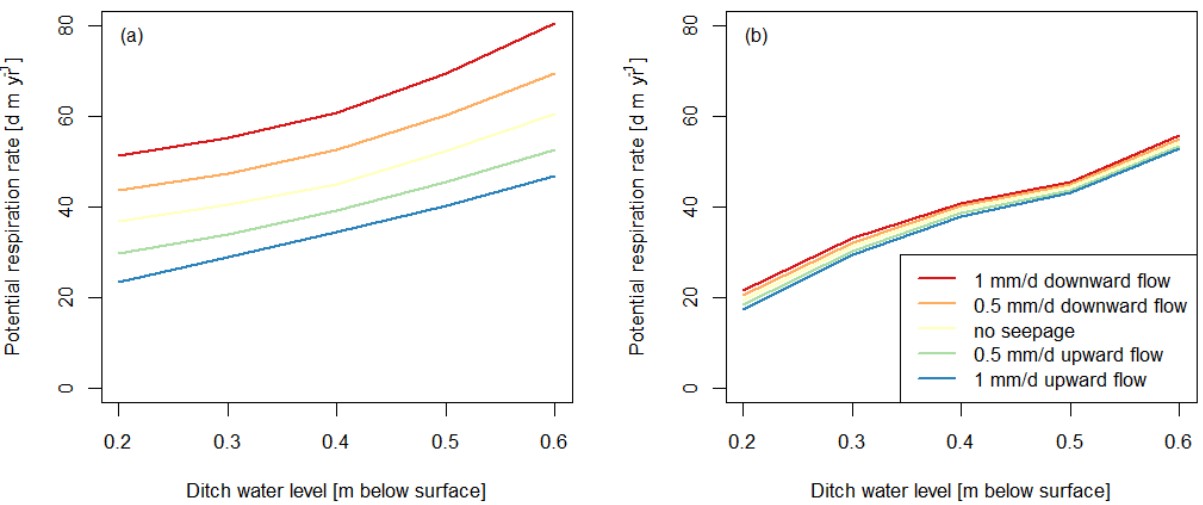

**Figure 9 – (a) The simulated effects of ditch water level on potential respiration rate for different seepage settings and (b) these
effects with SDSI in the dry year 2018. The colors of the lines represent the model seepage setting.**

### 3.5.3    Water table depth and estimated respiration rate

A frequently used proxy to estimate peat decomposition is mean groundwater table depth. In this section we present our
findings on the relation between simulated mean groundwater table depth and the NEP that was estimated with the modelled
respiration rate and Eq. 5.

We compared the mean modelled groundwater level with our simulated NEP and found a strong correlation between the two
variables (Fig. 10), in which summer (Eq. 6, $R^2 = 0.92$, May to August) performed better than annual (Eq. 8, $R^2 = 0.86$) mean
values. The slope and intercept of the linear regression models to estimate NEP with $WTD_S$ differed significantly for situations
with and without SDSI systems (Eqs. 6 & 7, $p < 0.05$). This means that similar water table depths lead to higher potential
respiration rate estimations when SDSI is applied (Fig. 10). To approach equal NEP for SDSI sites, the mean summer
groundwater level needs to be 10-15 cm closer to the surface than for non-SDSI sites. We see a clear difference between
simulated NEP of wet and dry SDSI model runs, which is due to SDSI drainage and irrigation dominance.

$$NEP_{control} = 6.55\,WTD_S - 0.99 \qquad \text{(Eq. 6)}$$

$$NEP_{SDSI} = 6.73\,WTD_S + 0.06 \qquad \text{(Eq. 7)}$$

$$NEP_{control} = 8.13\,WTD_A - 0.91 \qquad \text{(Eq. 8)}$$






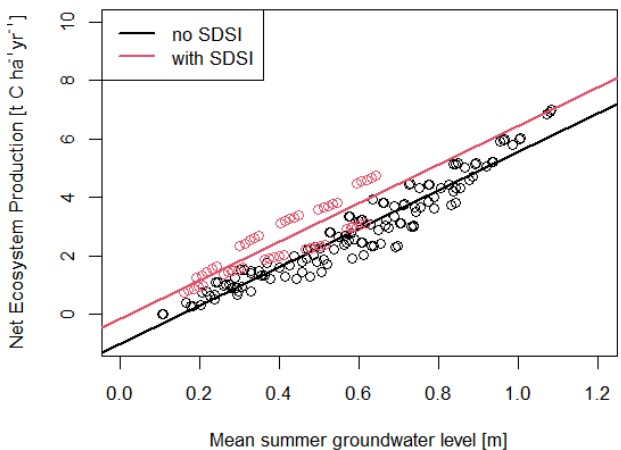

**Figure 10 –Mean modelled summer groundwater level and NEP for simulations with and without SDSI.**

## 4    Discussion

### 4.1    Observed effects of SDSI

Our measurements of water table depth show that SDSI maintained higher water tables during summer drought, as was found by Kechavarsi et al. (2007) and Querner et al. (2012). The effect was more evident for the pressurized SDSI system in Assendelft than for the passive SDSI in Vlist, where the pressure in the SDSI system followed the ditch water level. In Assendelft we also found more than 15% higher WFPS in the top 0.5m during July to September (Fig. 5e) compared to the control site. In Vlist the effects of SDSI on WFPS in the topsoil were not significant. However, we should be careful when interpreting the absolute values for WFPS or comparing WFPS-differences between Assendelft and Vlist due to high uncertainty in the relationship between sensor output signal and WFPS.

Our temperature measurements support the hypothesis that SDSI causes increased topsoil temperatures (to 0.5 m depth). In the summer, this effect was stronger in Assendelft than in Vlist, which is likely due to the pressurized SDSI in Assendelft. We hypothesize that the temperature increases were caused by the infiltration of warm ditch water and/or an increase in soil thermal conductivity by wetting the topsoil.

We found lower nocturnal $R_{eco}$ during summer for both plots with SDSI application suggesting that SDSI reduced peat respiration rate and/or crop productivity (Fig. 6b). This is in contradiction with the findings of Parmentier et al. (2009) which did not show a relation between groundwater table depth and $R_{eco}$ and Weideveld et al. (2021) who could not detect changes in $R_{eco}$ when SDSI was applied. In the autumn, when the SDSI systems switched from net irrigation to drainage, we expected



to find higher $R_{eco}$ on the SDSI parcels. Measured $R_{eco}$ in Assendelft supported this hypothesis, but in Vlist we did not find this effect (supplementary information, S2).

Our measurements show that the application of SDSI in 2020 reduced $CO_2$ emissions (NEP) by 83% ± 25% for Assendelft and 35% ± 17% for Vlist (Table 4). The pressurized SDSI of Assendelft thus resulted in higher emission reductions.

Measurements of C-export show that crop productivity was lower when SDSI was applied. For Assendelft and Vlist this decrease was 24% and 6% respectively, suggesting that there is a relation between groundwater table depth and crop production and/or between peat respiration rate and crop production. Although our results suggest vast benefits of applying SDSI, we have to note that the measurements have been done in the middle of the parcels and were taken in a severely dry year: conditions under which we expect maximum effect of SDSI. We expect considerably lower emission reductions for the complete parcel

area from ditch to ditch (35% and 39% for Assendelft and Vlist respectively) and over multiple years including years with less precipitation shortage. To estimate these effects we developed a model with which the potential respiration rate was calculated for the entire parcel and for both a meteorologically wet and dry year.

## 4.2     Simulated versus observed dynamics and SDSI effects

Generally, our model simulated the measured water table depths satisfactory. However, the simulated phreatic surface during summer did not show similar short term dynamics as observed on the research sites (supplementary information, S1). We think that this is a result of the complex characteristics of peat soils, such as the possibility to shrink and swell, preferential flowpaths, dual porosity and hysteresis, which were not represented with our model. We chose to use general parameter values based on metastudies (Liu, Janssen & Lennartz, 2016; Rezanzehad et al., 2016). The complexity challenges to describe water retention

and hydrology in the peat soil (Rezanezhad et al., 2016), as we experienced within this research. The hydrological model overestimated the water buffering role of the unsaturated zone during summertime. This resulted in a rather constant groundwater level unresponsive to summer drought or rainfall episodes, which were visible in the observed groundwater levels. To improve the simulation of groundwater table dynamics, several model experiments were executed such as incorporation of crack formation, increasing and decreasing residual water content and saturated water content, including hysteresis, adding

anisotropy or dual porosity. These attempts did not improve the simulated groundwater dynamics enough to warrant the extra complexity. The modelled average monthly groundwater table dynamics were in accordance with measured variables, as shown by Fig. 6b.

Temperature dynamics were simulated successfully, although simulated topsoil temperatures were slightly overestimated due to too high surface boundary temperatures. Our model predicted stronger topsoil warming with SDSI for Assendelft than for

Vlist, which was also measured. In Assendelft SDSI created a wetter topsoil than for Vlist. A wetter topsoil conducts heat better, which results in higher summer temperatures deeper in the profile.

On average 59% of the observed variation in $R_{eco}$ could be explained by simulated variations in WFPS and soil temperatures. This confirms our expectation that WFPS and soil temperature are dominant controls on microbial respiration dynamics, as stated by e.g. Mäkiranta et al., 2009 or Moyano et al., 2013. For the control parcels the variation in WFPS is dominantly a





seasonal variation related to deep groundwater tables in summer and high groundwater tables in winter and thus a strong cross-correlation exists between WFPS and soil temperature conditions. For the SDSI parcels the SDSI maintains a relative constant groundwater level and the observed variation in WFPS is much more rainfall and evaporation event driven, largely uncorrelated to the seasonal pattern of soil temperature. Therefore especially the SDSI sites provide an opportunity to unlink the activity curves for WFPS and soil temperature. We found that WFPS-activity curves that have a maximum activity around or below

0.65 WFPS and a general decrease towards zero respiration at 1.0 WFPS clearly outperformed other shapes of activity curves. In the literature relations were suggested that consisted of similar characteristics (Moyano et al., 2013), but measured relations vary widely and are far from general agreement (Kechavarzi et al., 2010; Moyano et al., 2013; Säurich et al., 2019).

Factors like microbial and plant growth, plant respiration, the influence of root exudates and organic matter quality are responsible for the relatively smaller differences between SDSI and non-SDSI $R_{eco}$ measurements (Fig. 6), as compared to the

differences between SDSI and non-SDSI modelled potential respiration rate. These factors dominate the absolute value of $R_{eco}$, whereas we only tested the part of the dynamics of $R_{eco}$ that can be explained by soil temperature and WFPS.

Our results show that our model captures effects of SDSI on water table depth, soil temperature and WFPS which translate to similar reductions in potential respiration rate as measured NEP reductions (Table 4). Therefore, we think that our straightforward approach to estimate water management strategy effectivity forms an alternative approach of intermediate

complexity compared to either groundwater table-GHG emission relationships (Fritz et al., 2017; Tiemeyer et al., 2020; Arets et al., 2020; Evans et al., 2021) or models that resolve the entire carbon cycle (van Huissteden et al., 2006; He et al., 2021). The additional complexity of unsaturated zone modelling of water flow and introducing soil temperature and a WFPS-activity curves requires a considerably higher modelling effort compared to approaches that rely on relationships between yearly average groundwater table and GHG-emissions, but it allows us to investigate the effect differences between groundwater

levels raised by ditch water level manipulations or by applying SDSI. Compared to models that describe the full carbon cycle and consist of many (often unknown) parameters, our approach forms a straightforward alternative to simulate the yearly carbon balance. Nevertheless, we should be careful when comparing emissions between locations as some location specific characteristics have not been simulated such as parcel width, organic matter content or C/N ratio.

**4.3    Model exploration of rewetting measures**

Our simulation results support the statement that there is an enormous potential to reduce greenhouse gas emissions from managed peatlands (Evans et al., 2021), which could be achieved by various water management strategies (Querner et al., 2012). Our results suggest that raising ditch water levels always results in decreasing potential respiration rates. Of course we can only state this for environments that are comparable with the situations we modelled and observed. Our results also show

that SDSI is not causing guaranteed decreases in potential respiration rate.

In fact, the complex interplay of the effects of SDSI on temperature and WFPS determine the total benefits of the systems. The most important effects that we encountered during the warm summer periods and in simulations without seepage fluxes are: (1) soil drainage during wet times increases potential respiration rate, (2) soil warming during dry times increases potential





respiration rate, (3) soil wetting during dry times decreases the depth over which respiration can occur but also (4) in some
cases increases respiration rate in the unsaturated topsoil as a result of constant moist conditions close to the warm soil surface
(Sect. 3.5.2). Overall, we see that the first and third effects are dominant when studying the final results. The second and fourth
effects are decreasing the overall effectivity of SDSI, but are less dominant. The specific hydrological seepage setting and
meteorology will determine which processes are active over the year and will determine long term effectivity of SDSI.

Our estimations of potential respiration rate for dry and wet years suggest that the extent of summer drought mainly determines
the effectivity of SDSI. The SDSI systems are likely to induce a higher amount of potential respiration rate during wet years,
and a lower amount of potential respiration rate during dry years – which is explained by the processes described in the previous
paragraph. As a result of climate change the occurrence of severely dry summers is more likely in the Netherlands (Philip et
al., 2020), inferring that SDSI leads to a net reduction of potential respiration rate on the longer term. Furthermore, the seepage
regime should always be taken into account when considering application of SDSI. It is likely that SDSI systems drain upward
seepage, which would normally support high water tables during drought, resulting in the increase of potential respiration rate
when applying SDSI under upward seepage conditions. Adversely, SDSI systems are likely to sustain groundwater tables in
situations with downward flow, resulting in a decrease of potential respiration rate when applying SDSI (Fig. 8b). Moreover,
in combination with elevated ditch water levels potential benefits of SDSI increase, which also suggests that the application
of pressurized drain subsurface irrigation tends to yield stronger decreases in potential respiration rate when drain pressure
exceeds ditch pressure.

Other research suggests a decrease peat decomposition (subsidence) by 45% when SDSI is applied during 2000-2001 (Querner
et al., 2012). Our results suggest that the effects of submerged drains would be within the range of these estimated SDSI effects
of dry and wet model scenario with conditions that apply for the research location: 0.5 mm d$^{-1}$ downward flow, a summer ditch
water level 0.3 m (and when observing the middle of the parcel).

Weideveld et al. (2021) found no clear benefits of SDSI in their setting. For model scenarios with low ditch water levels (60
cm below surface) we only predicted a slight decrease of 10% in potential peatland emissions over the full width of the parcel
for 2018. It is likely that moderate differences like these remained undetected in the study of Weideveld et al. (2021) as
uncertainty of NECB/NEP in almost all study fields exceeded 10%.

Besides effects of peatland management on aerobic respiration rate it is important to consider saturated-zone effects, like
infiltration of oxygenated and nutrient rich surface water, which we did not include in this research. As soil temperatures tend
to be increased with SDSI, also saturated respiration rate could be enhanced. Furthermore, SDSI enhances groundwater flow
and transport, possibly leading to higher nutrient and DOM leaching leading to adverse effects. Furthermore, often the pH of
the infiltrated surface water is higher than soil pH, enhancing respiration rate (Malik et al., 2018). Lastly, anaerobic respiration
(forming $CH_4$ and $N_2O$) is likely to increase when groundwater levels approach the soil surface, counteracting on the decreased
aerobic soil respiration in terms of GHG emissions (Tiemeyer et al., 2020).



### 4.4 Comparison with groundwater-GHG emission relations from previous studies

In many previous studies empirical relations have been established between land subsidence or NEP and (long term) groundwater characteristics like the yearly lowest, yearly mean groundwater levels, or mean summer groundwater levels (e.g. van den Akker et al., 2008; Fritz et al., 2017; Tiemeyer et al., 2020; Evans et al., 2021). In this study we fitted a linear model

between mean summer groundwater table/mean annual water table and our estimations of potential respiration rate. Our data suggest that peatland $CO_2$ emissions increase linearly with deeper annual groundwater levels, which is mostly in line with findings in literature except for the study Tiemeyer et al., 2020, who found an exponential relationship.

We compared our simulation results of the fitted relation between mean annual groundwater table and NEP with the findings of Fritz et al. (2017), Tiemeyer et al. (2020) and Evans et al. (2021)(Fig. 11, Table 5). From our modelling results, we derived

a relationship that closely follows the proposed relationship by Evans et al., (2021) with a slightly lower slope and a higher intercept. Our NEP estimations with high mean annual groundwater levels (< 0.10 m depth from surface) deviate from the confidence interval in the study of Evans et al. (2021), but are in accordance for deeper annual mean groundwater levels (Fig. 11). As our relation between groundwater depth and NEP is based on only four observed NEP values, the close match between our results and those of Evans et al (2021) forms a strong additional model validation. However, other studies of for example

Fritz et al. (2017) and Tiemeyer et al. (2020) generally suggest a higher NEP, especially for a $WTD_A$ exceeding 0.10 m in depth and more research is required to explain these differences.

**Table 5 –Overview of available fitted empirical relations to estimate NEP based on WTD$_A$.**

| Study | Function | Function with parameters |
|---|---|---|
| *Current* | $NEP = slope * WTDA + C$ | $NEP = 8.13\,WTDA - 0.91$ |
| *Fritz et al. (2017)* | $NEP = slope * WTDA + C$ | $NEP = 12.27\,WTDA - 0.02$ |
| *Tiemeyer et al. (2020)* | $NEP = CO_2(C)_{min} + CO_2(C)_{diff}\,e^{-ae^{bWTD_A}}$ | $NEP = -0.93 + 11.00\,e^{-7.52\,e^{12.97\,WTD_A}}$ |
| *Evans et al. (2021)* | $NEP = slope * WTDA + C$ | $NEP = 9.27\,WTDA - 1.69$ |

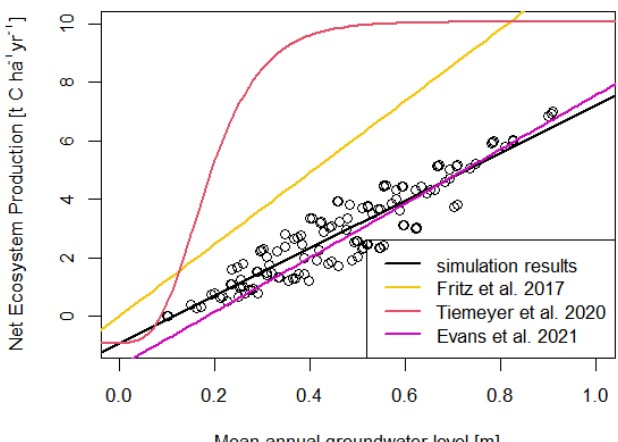





**Figure 11 - The relation between annual mean modelled groundwater level and NEP found in this study (Eq. 8) in comparison with the relation determined by Evans et al. (2021) containing all data of the complete boreal and temperate peatland measuring sites (Eq. 9).**

Mean annual water table depth can function as a predictor of peat decomposition (Tiemeyer et al., 2020; Evans et al., 2021),

although mean summer water table depth is found to be a more reliable predictor for predicting potential respiration rate in this research. Annual temperature fluctuations with high temperatures during summer and low during winter are an important driver for peat decomposition or potential respiration rate. Hence, the assumption that winter and summer water table depths are weighted equally to calculate NEP is incorrect. Therefore, we argue that methods that empirically relate groundwater table to peat decomposition can be improved by using summer water table depth.

We found that the relation between mean summer groundwater table depth and potential respiration rate diverges for simulations with and without SDSI, possibly because of the enhanced draining during wet periods and the topsoil-wetting and heat transporting effects of SDSI. Therefore, established relations on groundwater table depth and (approximations of) peat decomposition may not be applied to situations with SDSI (Fig. 10, Eq. 6, 7), and benefits of SDSI could easily be overestimated. For example, with a similar average summer water table depth of 0.6 m we find a higher respiration rate (25%)

for SDSI then for non-SDSI situations. As the water table depth becomes shallower, the relative difference increases even more (42% increase with a WTD of 0.3 m). Hence, the relations determined in other studies should be corrected when calculating the effects or $CO_2$ emissions of regions/parcels consisting of SDSI.

### 4.5 Implications for monitoring and rewetting strategies

As we found that our process-based approach estimating potential respiration rate as a measure for peat decomposition successfully captured the effects of rewetting measures on NEP, we would recommend to invest in measuring campaigns in managed peatlands that target to assess soil temperatures and WFPS. This will be an important addition to measuring groundwater table depth, as the effects of rewetting measures on potential respiration rate can be monitored more precisely. Particularly, the approach facilitates understanding the most important effects of SDSI on potential respiration rate. In fact, we

think that the concept of potential respiration rate could be used as a first step in determining a new standard for process-based quantification of (indicators of) peat decomposition (tier 3 approach, IPCC, 2014). We find that modelling studies are necessary when implementing this concept and when determining the effects of rewetting strategies, as we found that point measurements on the middle of a parcel are not representative for the total effects over the complete width of the parcel.

Following our results we find that rewetting policies in areas with low ditch water levels (>0.4 m below surface summer ditch

water level) should always include ditch water level increases. Investments in SDSI do not guarantee high reductions (>40%) in $CO_2$ emissions of managed peatlands over longer timescales when ditch water levels are low, and would not result in any $CO_2$ reductions when the region is prone to upward seepage that is higher than 0.5 mm d$^{-1}$. To assure high $CO_2$ emission reductions with SDSI, we advise to maintain high summer ditch water levels (0.2 m below surface preferably) or pressurized





SDSI. In the case of downward seepage, we would strongly advise to apply SDSI as a reduction in $CO_2$ emissions is guaranteed.

When summer groundwater table depth is lower than 0.2m from the surface methanogenesis and nitrate reduction are induced and this is likely to cause an offset in GHG reductions from the soil in dairy farming peatlands (e.g. Evans et al., 2021; Tiemeyer et al., 2020). Therefore, we would advise to keep summer ground water levels between 0.2 and 0.3m. By using pressurized SDSI these specific groundwater table depths could be targeted and maintained (as shown by the measurements in Assendelft). Quantification of the effects of parcel width was beyond the scope of our research as this would involve designing separate

model domains and analyzation methods which was not feasible within this project.

Finally, the water management strategies that we discuss are envisioned from a perspective in which agricultural business on managed peatlands could predominantly be maintained while reducing greenhouse gas emissions. However, several other strategies that embody the aim to reduce GHG emissions exist, such as full ecosystem restoration (Nugent et al., 2019) or paludiculture (Geurts et al., 2019). We emphasize the need for long term thinking and solutions and recommend

interdisciplinary research in which the broad goals of society as a whole are centralized while determining the fate of peatlands.

## 5 Conclusions

Field measurements of 2020 in Assendelft and Vlist show that SDSI systems tend to establish constant groundwater tables throughout the year, especially reducing the extent of the unsaturated zone in summer. Additionally, we found that SDSI also

tends to increase soils temperatures and WFPS in the upper soil zone (to 0.5 m depth). The effects were found to be stronger for pressurized SDSI systems (Assendelft). The yearly NEP reduction due to SDSI was $1.27 \pm 0.39$ kg $CO_2$ m$^{-2}$ yr$^{-1}$ ($83 \pm 25\%$) and $0.78 \pm 0.37$ kg $CO_2$ m$^{-2}$ yr$^{-1}$ ($35 \pm 17\%$) for Assendelft and Vlist respectively.

Model simulations showed similar seasonal dynamics as measured for groundwater table depth, soil temperature and WFPS. Also, the estimated effectivity of SDSI that was calculated using model output data of soil temperature and WFPS was in

accordance with measured effectivity for both field sites. This allowed us to extrapolate the measured effectivity at point-scale to an estimated effectivity for the entire field, reducing the effectivity of SDSI to 64% and 20% for Assendelft and Vlist respectively. Within the model exploration we found explicit effects of water management strategies on simulated potential aerobic respiration rate: (1) Elevating ditch water levels decreases potential respiration rate (up to 18 d m yr$^{-1}$ or 0.5 kg $CO_2$ m$^{-2}$ yr$^{-1}$), (2) SDSI is beneficial in decreasing aerobic potential respiration rate in environments with downward hydrological

flow or with drain pressure relatively close to the surface (up to 30 m d yr$^{-1}$ or 0.9 kg $CO_2$ m$^{-2}$ yr$^{-1}$) and (3) the effects of SDSI are more likely to be beneficial during severe summer drought. We found several effects of SDSI to be specifically important: (1) soil drainage during wet times increases respiration rate, (2) soil warming during dry times increases respiration rate and (3) soil wetting during dry times decreases respiration *depth* and at the same time (4) increases respiration rate in the unsaturated topsoil.

To approximate NEP and effectivities of rewetting efforts our approach yielded valuable results which were in line with previous scientific research in which mean annual groundwater level was used as a proxy for NEP. We do recommend using


a process-based approach that uses soil temperature and WFPS to estimate peat respiration rate, as it enables to understand and capture the fundamental effects of rewetting measures on potential respiration rate. If it is not possible to use soil temperature and WFPS, mean summer groundwater level instead of mean annual groundwater level should be used as a proxy

for NEP, since this proxy performed better and is a more logical choice as potential respiration rate is highest in warm soils. Furthermore, relations between groundwater table depth and NEP that have been determined without SDSI seemed to deviate from relations determined with SDSI, hence groundwater-respiration rate relations determined in the past may not be applied directly to calculate the effects of SDSI systems on NEP and have to be corrected.


**Data availability**

All raw data can be provided by the corresponding authors upon request.

**Author contribution**

JB, YvdV, MMH, KvH and GE outlined the research; CvH and JB performed the measurements; JB, YvdV, MvdB and RM

analyzed the data; JB and YvdV wrote the manuscript draft; MMH, MvdB, KvH, GE and RM reviewed and edited the manuscript.

**Competing interests**

The authors declare that they have no conflict of interest.

**Acknowledgements**

This study was part of the project "Nationaal onderzoeksprogramma broeikasgassen veenweiden" funded by the Dutch government to research greenhouse gas emissions emerging from peatlands. This project is an interdisciplinary collaboration between the following Dutch partners: STOWA, Deltares, Radboud Universiteit, Universiteit Utrecht, Wageningen Environmental Research, Wageningen Universiteit, Technische Universiteit Delft, B-ware, and Vrije Universiteit Amsterdam. We would like to thank all the researchers, technical staff and the farmers involved. Especially, we are grateful for the technical

support by Ron Lootens, Arie Bikker and Rob Stoevelaar of the VU.

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
