# Peer review of "Cutting peatland CO2 emissions with water management practices"

_Biogeosciences, 2021_

## Author Comment (AC1)

**Comments *Henk van Hardeveld* on manuscript 'Cutting peatland CO$_2$ emissions with rewetting measures (Biogeosciences discussion BG-2021-276)'**

*Introduction*

*We thank Henk van Hardeveld for his critical look and thoughtful comments that will improve the quality and readability our manuscript. Especially the comment on featuring more prominently our novel method to estimate peat respiration and the comment on highlighting the quantitative comparison with previous studies will both certainly raise the impact of our study. We are happy to apply revisions to improve our manuscript as formulated in the answers to the referee comments below.*

*RC1: 'Comment on bg-2021-276, line 68: Submerged drain subsurface irrigation (SDSI) systems', Henk van Hardeveld, 24 Nov 2021*

Throughout the years, these systems have been described in various ways. E.g., Querner et al. (2012) call them subsurface drains, Weideveld et al. (2021) call them subsoil irrigation and drainage systems, Hoogland et al. (2020; DOI:10.5194/piahs-382-747-2020) refer to them as drain infiltration, and Hoekstra et al. (2020; DOI: 10.5194/piahs-382-741-2020) favor pressurized drainage for a system similar to that on the Assendelft site. So why coin yet another name instead of using (parts of) a previous one, especially when the new name is less concise? Subsurface (and/or subsoil) seems superfluous: where else would the drains be? And submerged is not accurate all the times: a part of the appeal of these systems is that after a heavy rain shower, you can use them as conventional, non-submerged drains to more rapidly drain a field. Would (pressurized) drain irrigation systems not suffice?

*Reply on comment RC1:*

We acknowledge the need for the peatland community to establish a standardized term when referring to the technique that we called SDSI. In hindsight, we chose the term submerged drain subsurface irrigation (SDSI) as we thought this term would be most consistent, since it would indicate that drains remain *submerged* at all times and that the irrigation technique targets supplying water to the *subsurface* (subsoil) rather than the rooting zone which is conventionally targeted when discussing irrigation. We agree with your comment and think that it is important to use consistent terminology to avoid any confusion in the scientific debate and therefore, we will adhere to the term subsoil irrigation and drainage systems (SSI) that was used in Weideveld et al. (2021) and revise our manuscript accordingly. We believe that the term of Weideveld et al. (2021) is the best term available to describe the technique. As a matter of fact, when we chose the term SDSI the research of Weideveld et al. (2021) was not published yet.

*RC2: : 'Comment on bg-2021-276, line 95–96: aim', Henk van Hardeveld, 24 Nov 2021*

Two questions regarding your aim. First, a minor technical point, can you try to state your aim without using brackets? Surely, every part of your aim must by definition be important? Second, more importantly, can you try to allign your aim and your narrative more closely? I think the most important legacy of this paper will be that you introduce a novel method to more accurately assess the impacts of water management strategies on peat decomposition and greenhouse gase emission. So, must your new approach not take central stage? in your aim you mention various strategies, hydrological settings and meteorological conditions. But this strikes me as merely an afterthought. Once you have designed a better approach, by definition it will allow you to betrer explore the effectivity of strategies in different settings. It is nice that you do, don't get me wrong, but I think its is merely to demonstrate the added value of your approach.

In addition, please focus your Introduction on the processes that your approach addresses, avoid too much focus on anecdotal case studies such as you decribe in line 80–85, using vague phrases like "was suspected" and "the authors think". You might be aware that there has been much controversy about drain irrigation systems, sparked by a paper in bulletin 2018-06 of the International Mire Conservation Group. Arguably, the essence of this "knowledge war" is about a wide range in observed effectivities of these systems, and the question to what extent it is valid to use estimations based on water tables to estimate their effectivity. You method may help to settle this debate. For instance, in line 581–588 you make a strong point by using your method to explain why previous case studies in various settings come up with different conclusions.

Moreover, as your method might pave the way for better impact assessments, the comparison with previous methods should be better addressed in the Introduction section. I think Section 4.4 is one of the hightlights of your work, yet the previous methods are only discussed in very general terms in line 87–91.

*Reply on comment RC2:*

We were pleased to read that the novel process-based approach to estimate effectivity of peatland water management strategies is appreciated and are thankful for the comment that this approach should deserve more attention throughout the manuscript.

Based on this comment, we agree that our research aims can be presented more accurate and will revise our aim in the manuscript. The general aim is to measure, simulate and explore the effects that water management strategies may have on soil wetness and soil temperature and on the carbon balance, with emphasis on peat respiration. To achieve this aim, we presented the measurements and configured a model for an extended analysis. The novel process-based approach including soil moisture and temperature to simulate potential respiration indeed played a central role and therefore, we will elaborate upon this in the introduction of the revised manuscript.

We agree that we should limit the amount of discussion in the introduction to keep it concise and to promote readability and will rewrite this section

Thank you for the compliment that you consider Sect. 4.4 as one of the highlights of our work. The comparison provides confidence in the simulation results. We agree that the section is important for the international peatland community and will improve the introduction of this section in the revised manuscript.

*RC3: 'Comment on bg-2021-276: Section 2.2.3 and Fig. 4', Henk van Hardeveld, 24 Nov 2021*

I like this part of your approach, but please explain it more clearly. Fig. 4 is featured quite prominent, but the shapes seem random. I suspect this is not the case, that you have designed several categories. You merely state that they are "loosely based on the shape found by Säurich et al. (2019)". Can you elaborate on that? Especially because "the" shape of Säurich et al. (2019) does not exist. They present a wide variety of shapes and also mention that the variety would have been even bigger if they had included shapes found by other research.

*RC4: 'Comment on bg-2021-276: Results and Discussion', Henk van Hardeveld, 24 Nov 2021*

I strongly suggest that you analyze the sensitivity of your assessment.

Part of the controversy surrounding methods to assess the impacts of water management strategies in peatlands centers on their validity range. E.g., are methods derived on sites without drain infiltration systems also valid for sites with drain infiltration systems? If your method is to rise above such controversy, you cannot suffice by stating that your model simulates the water table dynamics "reasonably well" (line 318), or that the modelled temperatures were merely "slightly too high" (line 337).

According to the approach of Van den Akker et al. (2008), a 20 cm offset in the summer water table may cause up to 60% extra emission. And assuming a Q10 of 2–3, a 1.45 °C offset in temperature may cause a 10–17% increase in microbiological activity.

This raises the question to what extent you can accurately choose which WPFS optimum curve to use in your model? You have chosen shape 16, with a correlation of 0.591. But shape 8, which seems highly improbable has an almost similar correlation of 0.590.

Regardless of the results of your sensitivity analysis, I believe your approach will be a step forward compared to the current water table based approaches. But I do like to know just how robust your method is. Will a slight offset in your hydrological model or the chosen shape of the WPFS optimum curve produce similar, of very different results? And in case of high sensitivity, what is needed to accurately pinpoint which WPFS optimum curve to use? Multiple years of monitoring results on multiple sites, perhaps? In other words, are we there yet? Or are we merely still moving towards a better approach?

*Reply on comments RC3 and RC4:*

We thank you for the appraisal of Sect. 2.2.3. in which we discuss and present the tested curves that describe the relation between WFPS and potential respiration rate.

In the text we indeed refer to the shapes presented in figures in Säurich et al. (2019) and we agree that we need to be more specific on this. The WFPS-respiration shapes we refer to are based on Fig. 4a in the research article that includes $CO_2$-C emissions over a range of WFPS for fen and bog earthified topsoils.

The curves we tested (Fig. 4) were not random. In fact, we constructed a starting curve (curve 1) and tested the effect of changing four properties of the curve: the starting value of reparation rate at 1 WFPS, the shape of the curve (bèta distribution, normal distribution, linear), the (range of) WFPS value(s) with maximal potential respiration and (in case of distributions) the effect of standard

deviation or width. The testing results of the curves (comparison of Reco and the potential respiration rate calculated with the different curves) revealed curve properties that led to unsatisfactory correlations between $R_{eco}$ and potential respiration rate. We revealed that certain WFPS-relations and curve characteristics were invaluable which should be excluded. We think that we can improve Sect. 2.2.3 by describing and visualizing the structure we used while constructing the testing curves and by elaborating more upon our methodology.

We are pleased to read that you think that our approach including temperature and soil moisture conditions is a step forward compared to conventional water table based approaches and agree that the sensitivity of our approach needs to be tested. We will include a sensitivity analysis in our revised manuscript in which we test the effect of offsets in soil temperatures (similar test as changing the Q10 of our temperature-respiration rate curve), and the effect of the chosen WFPS-respiration rate curve, on the simulated effectivity for the model simulations that represent our research locations best. Within this analysis, we will exclude WFPS-respiration rate curves that produced unsatisfactory results. However, the seven curves that were performing well with a mean Pearson correlation > 0.55 in Table S2.1 will be tested with a sensitivity analysis.

We understand your point about the comparison between measurements and simulation results. The model was setup to represent a simplified version of a common managed peatland cross-section in the Netherlands. We chose one standard parcel width, implemented one simple soil profile consisting of only three horizons and did not implement vegetation growth and harvest. Hence, the model runs that we chose for the comparison with research sites were not based on specific field conditions, but the boundary conditions that we varied matched these boundary conditions as much as possible. Accordingly, the statements we make about the model performance must be placed in perspective. We think that the assumptions we made were realistic, and that the assumptions have similar consequences for each model run. The text could imply that we expect our model was specifically aligned with field conditions (line 317), this is however, not the case. We will clarify this in the introduction and methodology of the revised manuscript.

WFPS-respiration rate curve 8 indeed performed unexpectedly well. We think that decreases in respiration rate when WFPS < 0.65 were not significantly represented in the total simulated respiration rate. In Fig. S2.1, you can clearly see that the average WFPS in the top 30 cm of the soil profile will not drop below 0.65. Therefore, the effect of water shortage for microbes is rare in these model runs. However, this does not mean that conditions with a low WFPS that dampens microbial respiration activity do not occur. We know that the decrease in microbial respiration rate is still important for dry situations occurring in particular soil zones.

We think that more research is needed to improve the estimations on temperature- and WFPS-respiration curves. The perfect potential respiration curves will be difficult to define, as respiration is also influenced by dynamics in microbial soil communities, variations in decomposition state, soil aggregates, chemical status of the soil and management history. In our research we explore the potential of this approach, we gained trust in the application by various comparisons, but we cannot state that our relations between temperature/WFPS and potential respiration rate are perfect.

*RC5: 'Comment on bg-2021-276: Fig. 6', Henk van Hardeveld, 24 Nov 2021*

Can you elaborate on the potential respiration rate? How do you explain that Reco for the sites with and without irrigation drains are quite similar, but the potential respiration rate is much lower at the site with irrigation drains than at the control site? And how do you explain the sharp drop in potential respiration rate in September that is not matched at all by the measurements? It seems the drop can be related to high modelled water tables and the chosen WFPS optimum curve with zero activity at WFPS = 1. How do you interprete that?

Technical question regarding Fig. 6 (a): can you better explain which lines are plotted on which axes and add units to the third axis?

Suggestion: can you also show these graphs for the Vlist site?

*Reply on RC5:*

Thank you for your comment on potential respiration rate. The $R_{eco}$ that was measured includes peat respiration, plant respiration, respiration of fresh and easily degradable organic matter and anaerobic respiration. The potential respiration rate does only refer to peat respiration. We think that the water management strategies will mostly affect peat respiration and that the other forms of respiration are likely to only be slightly affected by strategies. That makes that the effect of the water management strategies that is reflected by $R_{eco}$ is buffered by these other components, and seems lower than the difference in potential respiration rate we simulated. We will highlight the differences between $R_{eco}$ and potential respiration rate in our revised manuscript.

The drop in potential respiration rate in September 2020 is indeed caused by the high modelled water tables and WFPS, and is a result of the WFPS optimum curve that we use. We think that the discrepancy between timing of the drop in measured $R_{eco}$ for the control parcel in Assendelft and the simulated potential respiration rate could be caused by a delay in the depletion of oxygen in the soil after saturation. Additionally, we see an increase in measured $R_{eco}$ for Assendelft control during the wet event in September that was not simulated. This could be a result of air that is pressed out of the soil during a large rewetting event, or other forms of respiration such as anaerobic respiration that was mentioned before.

The suggestion about Fig. 6 is very helpful: we will update Fig. 6 and the captions of the figure. A similar graph as in Fig. 6b can already be found for Vlist in the Supplementary information, Fig. S2.1, but was not included in the manuscript to reduce the length of the manuscript.

*RC6: 'Comment on bg-2021-276: Fig. 8', Henk van Hardeveld, 24 Nov 2021*

Arguably, the results for the Assendelft and Vlist sites are quite different. So it would seem pressurized irrigation drains and regular irrigation drains are two quite different systems, with different potential respiration rates in similar settings. Can you distinguish between both categories in the Figure? Currently, it is unclear which dots may refer to a pressurized system.

*Reply on RC6:*

The drain pressure in Assendelft was 30 cm higher than in Vlist and therefore, the differences we found in the results were expected. The selection of model runs did not specifically involve parcels with pressurized drainage, but we made the assumption that pressurized SDSI was represented by runs consisting of SDSI with higher ditch water levels as compared to the control situation. We understand that we should elaborate upon this in the manuscript and will do so in the revised manuscript.

*RC7: 'Comment on bg-2021-276: Section 4.4', Henk van Hardeveld, 24 Nov 2021*

The comparison between your approach and previous methods is very valuable. But the chosen relations seem a bit random. On the one hand, the relation of Fritz et al. (2017) was found in a semi-scientific Dutch magazine, which is only accessible by sending an e-mail to the authors. On the other hand, the often used relation of Couwenberg et al. (2011; doi.org/10.1007/s10750-011-0729-x) is lacking. As is the often used relation of Van den Akker et al. (2008), which uses the average summer water table, which in line 622 you claim is better than the average annual water table such as used by the chosen relations. Your comparisons will make a stronger point if you include those relations as well.

Technical comments: please explain what the dots in Fig. 11 are, change the units of the x axis of Fig. 11 and Fig. 10 into m below surface, and change the caption of Fig. 11 (in a concise manner) to match everything that you present.

*Reply on RC7:*

We are delighted to read your compliment on Sect. 4.4. We understand that the chosen relations that we compare with our results seem a bit random, especially the relation of Fritz et al. (2017). The relation of van den Akker et al. (2008) is not based on average groundwater levels but on ditch water levels and can therefore not be included. We agree that we should indeed include the model from Couwenberg et al. (2011) and might discard the relation of Fritz. et al (2017) within our revised manuscript.

The dots in Fig. 11 refer to model control simulations, we will update the legend and caption of Fig. 11. Furthermore, we will update the units on the x-axes of Fig. 10 and 11.

---

## Author Comment (AC2)

**Comments *Peatland Research* on manuscript 'Cutting peatland CO₂ emissions with rewetting measures (Biogeosciences discussions BG-2021-276)'**

Carbon fluxes from drained peatlands receive increasingly attention within various scientific disciplines. This paper follows this trend promoting an interdisciplinary approach. The authors have provided a valuable theoretical attempt to improve the community's understanding of soil moisture and carbon fluxes interactions. At the first sight the modelling work focuses on combining soil moisture, temperature and potential carbon mineralization rates for an improved quantification of hydrological variables steering seasonal peat losses.

However, after a second read through there is more to the paper. The authors incorporate a new method to approximate carbon fluxes from drained grasslands on peat quantitively. The new method relies on closed chamber technique. Chambers were supposed to close automatically 2-3 times per hour. The static chambers are reported to be surprisingly high (full 20 inches).

To compare the new chamber method with published data the authoring team builds a soil-water-carbon model. The 3.5 model exploration (Figure 1) helps to quantify how well the flux method can approximate existing carbon flux data at an annual resolution. Figure 11 highlights that the gas flux method deployed for model calibration in this study may systematically underestimate carbon fluxes from drained peatlands. The comparison with Evans et al. 2021 seems vulnerable given the almost absent overlap in grazing intensity and primary production of the sites included in both data sets.

The paper's modelling approach would need a proper cross validation with more established gas flux methods on the one hand and multi-year data sets for calibration and validation on the other hand. Multi-year carbon flux data sets are essential for quantifying main drivers of soil carbon, climate and water interactions in peatlands (e.g., https://doi.org/10.1111/j.1365-2486.2006.01292.x https://doi.org/10.1111/j.1365-2486.2009.02104.x ). More so where soil temperatures are likely to change methodically by static chambers that are commonly deployed for experimental warming at higher latitudes.

The title seems misleading. All 4 paddocks remained drained during the course of the experiment. 'Cutting peatland CO2 emissions with irrigation measures' would fit the content of the paper.

I enjoyed reading this version of the manuscript. Looking forward to updates of the model supported by cross-validation of carbon fluxes.

*Reply on RC8:*

We thank you for your critical reflection and the discussion points you brought up and are content to read that you evaluate the research as a valuable approach. We discussed your concerns with the team and are motivated to improve the manuscript as explained in the answer formulated below.

You mention that our method to estimate peatland carbon fluxes relies on the closed chamber technique. Our aim was to measure CO₂ fluxes with the least amount of soil and vegetation disturbance as possible. The height of the chambers is above the maximum vegetation height. Smaller chambers would not support the conditions that we find at the farmland. Furthermore, we are aware of affecting the microclimate of the soil and vegetation, with possible changes in air temperature, amount of wind, radiation, precipitation and air moisture content. Therefore, we chose not to compare model outcomes with the absolute observed CO₂ fluxes, but we chose to compare CO₂ flux-differences between different management regimes. Furthermore, we compared our

measured chamber ecosystem respiration dynamics with potential aerobic respiration rate dynamics that we calculated with a variety of WFPS-activity curves. We chose the WFPS-respiration activity curve that matched the dynamics the closest. An under or overestimation in chamber ecosystem respiration would not have any consequences for this comparison, as we solely rely on the daily and seasonal dynamics. We stimulate air mixing by using ventilators and change the location of the chambers every two weeks to limit the development of a micro-ecosystem and to achieve proper field representation. Our equipment has been tested in the lab and is calibrated each year.

We think that the chamber flux data that we used in our yearly carbon budgets give reliable estimates of the effects of different peatland management practices. Firstly, we found that our model supports our measured differences, as we found a similar reductions in yearly carbon budgets -that were constructed using the chamber measurements- as our model simulated for both our measuring locations. We did not calibrate our model on these differences but used literature and measured properties to describe soil water and temperature. Secondly, the research application of static automatic transparent chambers to measure greenhouse gas fluxes knows a long history and has been evaluated successfully frequently (Huth et al., 2017). Many published research articles are based on chamber datasets with highly limited measuring intervals and continuity (for example Görres et al., 2014; Tiemeyer et al., 2020). Interpolation is done with relatively simple light and temperature response curves, resulting in large uncertainties in the yearly carbon budget. In contrast, our temporal data coverage is very high (>90%) and interpolation is hardly needed. Following your comment, we stress reliability of transparent chamber measurements by referring to research articles in which comparable chamber methodologies were used to quantify $CO_2$ fluxes with similar vegetation settings within our revised manuscript.

Indeed, it would be great to be able to present a multiyear cross-validation between chamber and eddy-covariance measurements. However, it has been already been proven that eddy-covariance and chamber measurements yield comparable results (Frolking et al., 1998; Laine et al., 2006; Stoy et al., 2013). Besides this and the arguments that we provided earlier -to explain why we can rely on the chamber measurements- the peatland community is in need of knowledge on how to prevent greenhouse gas emissions from managed peatlands. We are currently processing eddy covariance and chamber data of 2021, and plan to publish the outcomes of the comparison. Nevertheless, this should not constrain the publication of this research article. As a matter of fact, eddy covariance also induces many uncertainties (affected by choices in measurement set-up and methodologies for analysis).

We regret to read that our title could be misleading and will consider alternative options for the term *rewetting*. However, only referring to *irrigation* measures in the title as you suggest would miss the ditch water level elevation measure to reduce peat respiration.

We agree upon the fact that extensively used grasslands are underrepresented in the research of Evans et al. (2021). However, the authors state that annual groundwater levels "override other ecosystem- and management-related controls on greenhouse gas fluxes". Therefore, we think that the comparison with Evans et al. (2021) within our research is appropriate. Nevertheless, we included other important relations between annual water table depth and $CO_2$ emissions. Within our revised manuscript, Figure 11 will be updated with the relation from Couwenberg et al. (2011). Following your comment we will consider in text comparisons featuring the other relations plotted in Fig. 11 instead of highlighting the comparison with Evans et al. (2021).

**References**

Frolking, S. E., Bubier, J. L., Moore, T. R., Ball, T., Bellisario, L. M., Bhardwaj, A., Carroll, P., Crill, P. M., Lafleur, P. M., McCaughey, J. H., Roulet, N. T., Suyker, A. E., Verma, S. B., Waddington, J. M., and Whiting, G. J.: Relationship between ecosystem productivity and photosynthetically active radiation for northern peatlands, Global Biogeochem. Cycles, 12, 115–126, https://doi.org/10.1029/97GB03367, 1998.

Görres, C. M., Kutzbach, L., and Elsgaard, L.: Comparative modeling of annual CO2 flux of temperate peat soils under permanent grassland management, Agric. Ecosyst. Environ., 186, 64–76, https://doi.org/10.1016/j.agee.2014.01.014, 2014.

Huth, V., Vaidya, S., Hoffmann, M., Jurisch, N., Günther, A., Gundlach, L., Hagemann, U., Elsgaard, L., and Augustin, J.: Divergent NEE balances from manual-chamber CO2 fluxes linked to different measurement and gap-filling strategies: A source for uncertainty of estimated terrestrial C sources and sinks?, Zeitschrift fur Pflanzenernahrung und Bodenkd., 180, 302–315, https://doi.org/10.1002/jpln.201600493, 2017.

Laine, A., Sottocornola, M., Kiely, G., Byrne, K. A., Wilson, D., and Tuittila, E. S.: Estimating net ecosystem exchange in a patterned ecosystem: Example from blanket bog, Agric. For. Meteorol., 138, 231–243, https://doi.org/10.1016/j.agrformet.2006.05.005, 2006.

Stoy, P., Williams, M., Evans, J., Prieto-Blanco, A., Disney, M., Hill, T., Ward, H., Wade, T., and Street, L.: Upscaling tundra CO2 exchange from chamber to eddy covariance tower, Arctic, Antarct. Alp. Res., 45, 275–284, https://doi.org/10.1657/1938-4246-45.2.275, 2013.

Tiemeyer, B., Freibauer, A., Borraz, E. A., Augustin, J., Bechtold, M., Beetz, S., Beyer, C., Ebli, M., Eickenscheidt, T., Fiedler, S., Förster, C., Gensior, A., Giebels, M., Glatzel, S., Heinichen, J., Hoffmann, M., Höper, H., Jurasinski, G., Laggner, A., Leiber-Sauheitl, K., Peichl-Brak, M., and Drösler, M.: A new methodology for organic soils in national greenhouse gas inventories: Data synthesis, derivation and application, Ecol. Indic., 109, 105838, https://doi.org/10.1016/j.ecolind.2019.105838, 2020.

---

## Author Response (AR1)

**First round of major revisions that were applied for "Cutting peatland $CO_2$ emissions with water management practices" – Biogeosciences Discussions**

We thank the associate editor for the opportunity to revise our manuscript and both reviewers for their thorough comments and valuable suggestions. We have extensively revised our manuscript based on the reviewers suggestions with several new and improved results. The most important ones are listed here:

- We changed the title of our manuscript, as the a member of the peatland community anonymously suggested. *Rewetting measures* might be interpreted a full ecosystem restoration, instead, we focus on *water management practices* within our manuscript.
- We decided to process and add the data of measuring year 2021 to our manuscript. We expected that this would make the manuscript more robust, suggestions of both reviewers that multiple years of monitoring data could help improving the examination of our methodology. We do understand that the reviewing-process could be delayed by this addition, but in our opinion this is justified by the improved quality and potency of the manuscript.
- We changed Net Ecosystem Production (NEP) to Net Ecosystem Carbon Balance (NECB) as this definition is more accurate. Also, we changed the term *submerged subsurface irrigation systems* (SDSI) to *subsoil irrigation and drainage systems* (SSI) as suggested by reviewer 1.
- The formatting of the references was inconsistent with the Copernicus formatting. We corrected for this.
- We discovered an error in the calculation of C-export through harvest. We updated the calculations, outcomes. Also, the Supplementary information was updated and the data sheet has been restructured.
- We improved our gapfilling method and error estimation for chamber data. This slightly changed measured NEE and NECB.
- We simplified the visualization of $R_{eco}$ and potential respiration rate dynamics in Fig. 6 and included Vlist as reviewer 1 suggested.
- The sand soil around drain tubes was not represented correctly in our model. This had consequences for the potential respiration rate in simulations with SSI. Results have been recalculated for SSI scenarios and figures and text were changed accordingly. Especially the outcome of Fig. 10 was affected by this error. Previously we found that the intercept of the estimated linear relations between mean summer water table depth and NECB differed for control and SSI simulations. However, now we find that the slope estimates differ instead of the intercept estimates. We further improved Fig. 10 by adding an extra subplot that shows the differences between SSI and control NECB simulations during a dry year. To present the linear models that were fitted, we inserted a table that also replaces Eq. 6, 7 and 8.
- Reviewer 1 indicated that our simulation outcomes might be sensitive to the WFPS and temperature curves that were used. This was especially the case for the WFPS curves, and we are happy to present our sensitivity analysis in Sect. 2.2.3, 3.4.1. and Supplementary information S2.
- We adjusted Fig. 7 and now also show the dependency of measured NECB on mean summer and annual groundwater levels.
- We elaborated upon various aspects of our research that were unclear to the reviewers and the community:
    - The aim of the research in Sect. 1.
    - The effects of a 'micro-climate' induced by the automatic transparent chambers in Sect. 2.1.4.
    - The extent to which we aligned our model simulations with the field sites in Sect. 2.2.

- o The definition of potential respiration rate and how this concept should be utilized in Sect. 2.2.2.
- o The comparison between potential respiration rate and $R_{eco}$ in Sect. 2.2.3.
- o The representation of pressurized SSI within our model in Sect. 2.2.4.
- o The implications of our WFPS curve selection in Sect. 4.2.
- We improved axis labels of multiple figures as suggested by reviewer 1.
- As reviewer 1 suggested, we updated Fig 11. with the relation of Couwenberg et al., (2011). In Sect. 4.4. we also avoided a direct in-text comparison with Evans et al., (2021), following the suggestion of the peatland community reviewer, as extensive grasslands were indeed underrepresented in the study.
- We improved the conclusion by centralizing the aim of the article and avoiding repetition of results.

---

## Author Response (AR2)

**Second round of revisions that were applied for "Cutting peatland CO$_2$ emissions with water management practices" – Biogeosciences Discussions**

We thank the associate editor for his decision to proceed with the discussion on our manuscript. Also, we thank both reviewers for their feedback on the article. We have already responded to reviewer 1, who recommended our manuscript for publication, and adapted our manuscript accordingly. Here, we specifically address the feedback of reviewer 2 who was less positive and recommended rejection. First, we respond to the comments, then we address the changes we applied in this second revision.

The main criticism of reviewer 2 is that the modelled daily potential respiration rate cannot be compared with daily measured fluxes, as the short-term carbon cycle and vegetation dynamics are not represented by the model. This is indeed a very fundamental point, but to us it also shows that reviewer 2 did not fully grasp our approach. In our manuscript we use the measured R$_{eco}$ twice.

1) For each of the 2 years and four sites we add up all daily R$_{eco}$, GPP and harvests to derive a yearly estimate of the carbon oxidized from the peat (NECB). We cut up the individual years (i.e., at January 1$^{st}$) and assume that differences in the short-term carbon stocks (grass, roots soils) between years are small compared to the year-round carbon oxidized from the peat and the uncertainty thereof. Using measured yearly NECB as a best estimate for the loss of carbon from peat is a very common approach. This has also been done in Evans et al. (2021) and in Tiemeyer et al. (2020). The model introduced in this paper was validated on NECB of two years (a dry and wet year) from four different fields. Relative differences in yearly averaged potential respiration rate and measured NECB correspond very well (Fig. 7). Hence, this supports the assumption that the differences in storage of young carbon cycle material at the start and end of a measuring/modelling year are small compared to the uncertainty in NEE and harvests (Sect. 2.1.4).

2) We do compare measured daily R$_{eco}$ with computed daily potential respiration rate, but the goal of this comparison is only to compare the dynamics of both. We test several water-filled-pore-space (WFPS)-sensitivity curves (relation between water content and potential respiration) that fit the general shape shown from lab measurement by Säurich et al. (2019). The curve that explains most of the observed daily dynamics of R$_{eco}$ is assumed to best represent the effect of soil moisture on respiration, where we indeed cannot distinguish between fast and slow carbon cycle respiration and thus assume the same sensitivity for all carbon pools, similar to the approach of van Huissteden et al. (2006). If our aim would have been to simulate short term ecosystem respiration (R$_{eco}$), it would indeed matter to account for short cycle carbon, as this would have influenced the magnitude of the flux. However, we only used the R$_{eco}$ *dynamics* to establish the effect of WFPS on potential respiration. Even though short-cycle carbon respiration disturbs the comparison between R$_{eco}$ and potential respiration we find high correlations. We know that the shape of the curve is bound by experiments in literature, and we also apply a sensitivity analysis to support the choice for our WFPS-potential respiration curve. Therefore, we think that our approach is actually a step forward in addressing the peat respiration sensitivity for soil moisture under field conditions.

In short: In contrast to what reviewer 2 states, we do not use modelled daily potential respiration rates to represent measured daily R$_{eco}$ anywhere in our manuscript. Based on the comments of reviewer 2, we do now make another effort to make this distinction extra clear within the manuscript.

When designing the research, we aimed for a model that consists of a low amount of parameters. Indeed, the young carbon pool could be modelled, as was done in some of the articles that reviewer 2

suggests, but adding complexity to models is not inherently leading to better model results due to a higher amount of parameters. The added complexity might induce problems such as equifinality and increased uncertainty due to parameter estimation.

It seems that reviewer 2 addressed a mistake that was made while converting the potential respiration to NECB in line 482 (Sect. 3.5.1). We thank reviewer 2 for noticing this. However, by definition of our approach (relating the modelled yearly potential respiration to the observed yearly NECB with a linear conversion) there is no bias between modelled and observed NECB (Eq. 5).

Reviewer 2 argues that other vegetation types would expose the effect of the short-term carbon cycle. Indeed, we think that a different vegetation class would change the magnitude of $R_{eco}$. However, a natural system would also have different soil/peat characteristics that influence the decomposition. We specifically studied drained agricultural peat meadows within this research and natural systems were beyond our study scope.

The reviewer argues that without modelling C-input it is not possible to obtain changes in the C-stock of peatlands. We disagree with his argumentation. The fact that the observed NECB and the modelled yearly potential respiration rate align so well indicates that indeed our approach has great potential for simulating yearly decomposition fluxes from agricultural drained peat soils. Also, the assumption of differences in short-term carbon stocks between the start and end of the measuring/modelling years does not seem to affect our results given the high rates of peat decomposition found in these agricultural drained peatlands. Thus, there is no reason for suspecting large errors or flux imbalances.

**Changes**

- We elaborated upon the measurement and modelling assumptions related to the short carbon cycle within Sect. 2.1.4, 2.2.2, 2.2.3, 3.4.1.
- We elaborated upon the differences between the comparison of $R_{eco}$ and potential respiration rate, and the comparison of the NECB (including Reco) and yearly potential respiration rate within Sect. 2.2.3.
- We corrected for the conversion error in the first sentence of Sect. 3.5.1 and checked other conversions throughout the article.
- We corrected for two typing errors in Eq. 5: the '±' sign should have been a '+' and the intercept *0.269* should have been *0.259*.

---

## Author Response (AR3)

We thank the editor for giving us opportunity to respond to the concerns of reviewer 3. Of course, we are disappointed to receive such a critical review with few constructive comments to improve our manuscript. We do not agree with the evaluation of reviewer as we describe below.

The motive of the work is misguided. As stated in the abstract, "the effects of rewetting efforts on microbial respiration rate are largely unknown". This statement does not consider extensive work in California, USA which has quantified the effects of rewetting of peat on GHG emissions and NECB (e.g. Hemes et al. 2019).

- In general, literature indeed reports that wetter conditions tend to lower CO2 emissions and increase methane emissions. However, rewetting by raising ditch/surface water levels, periodic inundations, or by subsurface irrigation will affect water table, soil moisture and soil temperature differently. Consequently, microbial respiration will also be affected differently for each of these rewetting measures. This causes for example that currently there is no consensus on the effects of SSI (as described in Sect. 1, line 73). This is why we state that *"the effects of rewetting efforts on microbial respiration rate are largely unknown"*.
- We found that Hemes et al. (2019) included only one drained agricultural peat grassland site that was not representable for the situation of the peatlands that we describe in our article as this site was intermittently inundated and located in a much warmer climate. Furthermore, groundwater tables and soil moisture content were not measured within the research making it impossible to quantify the effects of rewetting on soil hydrology itself. The literature that was used within our manuscript gives an accurate representation of the available knowledge on peat decomposition in managed agricultural peat grasslands.
- We can, however, improve referencing to non-EU literature to illustrate that agricultural peat decomposition is globally occurring and that it is relevant to improve process understanding and knowledge on mitigation strategies.

Second, the results of this study do not add significantly add to the scientific literature related to the effect of peat rewetting. As stated in the article numerous authors have demonstrated the GHG emissions reduction associated with peat rewetting.

- Our scientific advances are: We present (1) new data on the effects of SSI, a (2) novel methodology to simulate peat decomposition (and CO2 emission-reduction) in which soil moisture and soil temperature control microbial respiration and spatial differences within agricultural fields are accounted for (2D profile) and (3) show that spatial differences in CO2 emission-reduction after implementing measures can be understood and modeled by site specific hydrology and meteorology over time. Apart from annual WTD, no other drivers or field boundary conditions (like seepage, meteorology, soil temperature or soil moisture) have been investigated in literature.
- We qualified and quantified the effects of SSI on soil respiration, which has not been done by previous researchers. These effects (qualitatively and quantitively) have largely been unknown. Therefore, we do not understand and do not agree with the point of reviewer 3 that we do not add significant knowledge.

Third, the methods and modeling are problematic and do not support the statement that "Our findings can contribute to peatland management, to better decide on where and how water management practices would be effective". Technical problems include the following.

- The aim of our modelling is to show that with nationally available soil datasets we can describe groundwater levels, soil moisture, soil temperature, CO2 emissions and reduction of these emissions by SSI reasonably well. Well enough to 1) explore the effect of environmental conditions such as seepage, weather and soil type of the reduction of CO2 emissions by two types of rewetting: raising the surface water level and subsurface irrigation, and to 2) recommend this modelling-method to improve our estimations of countrywide emissions from peatland that are currently still based on landcover specific emission factors and universal water table emission relationships (Sect 1, line 89).
- Groundwater modeling results show poor agreement with measured values.
  - Measuring and modelling groundwater table depth in a peat soil is a challenge especially due to the high saturated water content and thick capillary fringe (substantial differences in groundwater table may correspond to small differences in air-filled pores space relevant for decomposition), surface movement due to swell and shrinkage and preferential flows likely to occur within dry summers when peat shrinkage imposes preferential flow paths. We did elaborate upon the groundwater modelling results in Sect. 4.2. In our opinion the results were satisfactory.
- Water filled pore space modeling results show poor agreement with measured values.
  - Seasonal patterns in soil moisture are generally well described by the model. However, absolute water content, and absolute changes in water content are typically difficult to measure accurately in peat soils. Soil hysteresis and the high saturated water contents (>85%) causes that there is no strong relationship between soil moisture sensor values and water content measurements. The measurements do show a strong seasonal signal that relates to the air-filled pore-space relevant for decomposition
  - Drying and wetting hysteresis due to peat shrinking and swelling and soil heterogeneity over depth introduce serious challenges when interpreting WFPS measurements. Currently, we are working on optimization of sensor data interpretation by including tensiometer datasets.
  - We know that peat may need several months to fully swell and re-wet, while the model may indicate that the soil is completely saturated in early autumn (as was the case in Assendelft as described in Sect. 3.3). At present, no alternative theory or model exists to deal with these particular hydrological properties of peat (Sect. 4.2).

• No input model parameters such as hydraulic conductivity values for the hydrologic model were measured. The lack of measurement of input values resulted in a ill-conditioned model.

- When developing our model no measurements of hydraulic conductivity or water retention characteristics were available. Therefore, we decided to design a model based on general characteristics of a drained peat soil. Unlike reviewer 3 we think that the lack of site-specific input values is an advantage of our methodology, as it makes our results broadly applicable to a variety of agriculturally drained peat soils. Our results support this claim.

• The simulations do not account for varying soil carbon contents which have been shown to affect CO2 emissions and subsidence (e.g. Deverel et al. 2016).

- The soil organic matter (OM) density within our peat soils is very similar between the four measuring plots (15.64, 14.81, 14.82, 14.40 g OM cm-3 for Vlist control, Vlist SSI, Assendelft control, Assendelft SSI, resp. in the top 1.2 m of the profile). When implementing measures on a particular site we know that the soil, and soil organic matter content, will remain similar. Indeed, site characteristics may have an effect on the magnitude of the flux, but not on the effectivity of a certain water management measure. We clarified this within the revised version of our manuscript.
- The research of Deverel et al. (2016) includes all soils within a delta area, including shallow peat soils (< 1 m thick) and low organic matter content soils. In our research locations, soil organic matter content is much less variable and our results only apply to thick peat soils (>1.5 m of peat).

• Methane emissions were not accounted for in the NECB calculations. These have been shown to be significant in drained peat soils (e.g. Glenn et al., 1993, Anthony and Silva, 2021) and pasture on organic soil (e.g. Hatala et al. 2012).

• Nitrous oxide emissions were also not considered. These have been shown to be significant in drained peat soils (e.g. Anthony and Silver, 2021).

- It was never the aim of our research to make a net ecosystem greenhouse gas balance, but a carbon balance to estimate the CO2 emissions from peat oxidation. There will be a small part of the carbon emitted as CH4, but this is insignificant compared to the carbon emitted by CO2, which was shown in unpublished CH4 data from a previous study from the Assendelft research plots (including the SSI plot with highest groundwater levels).
- Quantifying methane and nitrous oxide emissions was beyond the scope of this research and we mention that groundwater levels that approach the surface may induce  $CH_4$  or  $N_2O$  emissions (Sect. 4.3) and therefore suggest to keep groundwater levels below -0.2 m (Line 740, Sect 4.5).
- Nevertheless, methane and nitrous oxide emissions can indeed be significant when determining the total greenhouse gas budget, especially when soil inundation occurs (as described in Anthony and Silver, (2021); Hemes et al. (2019), which is not the case at our research locations. Hatala et al. (2012) found that most CH4 emissions emerge from ditches. It would be interesting to measure CO2, CH4 and N2O emissions in future research when aiming for complete greenhouse gas budgets (again this is not similar to NECB).
- To avoid any confusion in the future we checked the manuscript for misleading usage of the term *greenhouse gas emissions* within the revised version of the article.

Lastly, because of these issues, I conclude that the study does not provides a greater "process-based understanding in these rewetting effects on peat decomposition". And I disagree that there has been a successful "integrating of high quality field measurements and literature relationships with an advanced hydrological modelling approach.".

Still, reviewer 3 did not present alternatives for an improved process-based understanding of peat decomposition or alternatives to quantify the effects of water management strategies (except expensive measurement protocols). We regret to read that reviewer 3 neither mentions anything about our findings on the effects of SSI, our reasoning to focus on water/air filled pores and soil temperature, nor the measured yearly carbon balances which were successfully reproduced in our simulations. We remain very content with the outcomes that we could realize.

**References**

Anthony, Tyler L. | Whendee L. Silver, 2021, Hot moments drive extreme nitrous oxide and methane emissions from agricultural peatlands, Global Change Biology, DOI: 10.1111/gcb.15802

Deverel, S. J., Ingrum, T., and Leighton, D. A.: Present-day Oxidative Subsidence of Organic Soils and Mitigation in the Sacramento-San Joaquin Delta, California, USA, Hydrogeol. J., 24, 569–586, https://doi.org/10.1007/s10040-016-1391- 1, 2016.

Glenn, Shannon Andrew Heyes, Tim Moore, 1993, Carbon dioxide and methane fluxes from drained peat soils, southern Quebec, Global Biogeochemical Cycles Volume 7, Issue 2 p. 247-257,

Hatalaa Jaclyn A.,\*, Matteo Dettoa,b, Oliver Sonnentaga,c, Steven J. Devereld, Joseph Verfaillie a, Dennis D. Baldocchi. 2012, Greenhouse gas (CO2, CH4, H2O) fluxes from drained and flooded agricultural peatlands in the Sacramento-San Joaquin Delta, Agriculture, Ecosystems and Environment, 150 (2012) 1– 18

Hemes, K. S., Chamberlain, S. D., Eichelmann, E., Anthony, T., Valacha, A., Kasaka, K., Kuno, S., Daphne, V., Joe, S., Whendee, L., and Baldocch, D. D.: Assessing the Carbon and Climate Benefit of Restoring Degraded Agricultural Peat Soils to Managed Wetlands, Agr. Forest Meteorol., 268, 202–214, 2019.

---

## Author Response (AR4)

**Authors response file upload**

We thank the editor for his acceptance of the manuscript and the outstanding communication during the review-phase. The long review process has been difficult as a young first-author, but thankfully the editor gave us the opportunity to respond to all reviewer concerns – and invested much time to evaluate the discussion.

*Changes in the final manuscript and supplement:*

- Supplement material was re-numbered as requested by Polina Shvedko.
- We checked the orders of authors in the manuscript and MS records and found that these are identical (as requested by Polina Shvedko).
- We deleted an error in line 469. We refer only to harvest statistics in the supplements, not NEE.